# Weighted Clock Logic Point Process

Ruixuan Yan[1], Yunshi Wen[1], Debarun Bhattacharjya[2], Ronny Luss[2], Tengfei Ma[2],
and Achille Fokoue[2], and Agung Julius[1]

[1]Rensselaer Polytechnic Institute
[2]IBM T.J. Watson Research Center

## Abstract

Datasets involving multivariate event streams are prevalent in numerous applications. We present a novel framework for modeling temporal point processes called clock logic neural networks (CLNN) which learn weighted clock logic (wCL) formulas as interpretable temporal rules by which some events promote or inhibit other events. Specifically, CLNN models temporal relations between events using conditional intensity rates informed by a set of wCL formulas, which are more expressive than related prior work. Unlike conventional approaches of searching for generative rules through expensive combinatorial optimization, we design smooth activation functions for components of wCL formulas that enable a continuous relaxation of the discrete search space and efficient learning of wCL formulas using gradient-based methods. Experiments on synthetic datasets manifest our model's ability to recover the ground-truth rules and improve computational efficiency. In addition, experiments on real-world datasets show that our models perform competitively when compared with state-of-the-art models.

## 1 Introduction and Related Work

Multivariate event streams are emerging types of data that involve occurrences of different types of events in continuous time. Event streams are observed in a wide range of applications, including but not limited to finance (Bacry et al., 2015), politics (O'Brien, 2010), system maintenance (Gunawardana et al., 2011), healthcare (Weiss & Page, 2013), and social networks (Farajtabar et al., 2015). As opposed to time series data that typically comprises continuous-valued variables evolving in regular discrete time stamps, event streams involve events occurring irregularly and asynchronously in continuous time. Modeling the dynamics in event streams is important for a wide range of scientific and industrial processes, such as predicting the occurrence of events of interest or understanding why some deleterious events occur so as to possibly prevent their occurrence. A *(multivariate) temporal point process* (TPP) provides a formal mathematical framework for representing event streams, where a *conditional intensity rate* for each event measures its occurrence rate at any time given the historical events in the stream (Daley & Vere-Jones, 2003; Aalen et al., 2008).

There has been a proliferation of research around TPPs in recent years, particularly around the use of neural networks for modeling conditional intensity rates as a function of historical occurrences (Du et al., 2016; Mei & Eisner, 2017; Xiao et al., 2017; Xu et al., 2017; Gao et al., 2020; Zhang et al., 2020; Zuo et al., 2020). One stream of research studies *graphical event models (GEMs)* as a compact and interpretable graphical representation for TPPs, where the conditional intensity rate for any particular event depends only on the history of a subset of the events (Didelez, 2008; Gunawardana & Meek, 2016). While any TPP can be represented as a GEM, various models make assumptions about the parametric form of conditional intensity rates for the sake of learnability, for instance that rates are piece-wise constant with respect to occurrences within historical windows (Gunawardana et al., 2011; Bhattacharjya et al., 2018). *Ordinal GEMs(OGEM)* (Bhattacharjya et al., 2020; 2021) are a recent model from this family where a conditional intensity rate depends on the order in which parent events occur within the most recent historical time period.

A temporal logic point process (TLPP) framework was proposed as an alternate way to lend some interpretability to TPPs by modeling intensity rates using temporal logic rules (Li et al., 2020). Although the initial work pre-specified temporal logic rules, recent work has introduced a temporal logic rule learner (TELLER) for automatically discovering rules (Li et al., 2021). There is however

the issue of scalability since TELLER exploits an expensive branch-and-price algorithm to search for temporal logic rules in a discrete space. Another important limitation of this work is that TELLER's rules are not informative enough to explain how the interval length between ordered events impacts the conditional intensity rate. For instance, while predicting the occurrence of diabetes, the rule that "insulin injection happens 20 minutes before eating meal" is more informative and accurate in predicting "blood glucose remains normal" than the rule that "insulin injection happens before eating meal", as the latter rule cannot expose the interval between 'insulin injection' and 'eating meal'. To tackle the above limitations, we propose novel *atomic predicates* enriching the expressiveness of temporal logic rules as well as a differentiable framework to learn rules in an end-to-end manner.

This work introduces a differentiable neuro-symbolic framework, *clock logic neural network (CLNN)*, to model TPPs by learning *weighted clock logic (wCL)* formulas as explanations. Firstly, event streams are converted into continuous-time *clock signals* representing the time interval between the last occurrence of an event and the current time. Next, we propose a novel wCL to describe the underlying temporal relations with relative interval length, enabling the design of a CLNN to learn the generative mechanisms. Instead of searching for temporal logic rules in some vast discrete space, CLNN associates every neuron with an order representation or a logical operator and assigns weights to edges to reflect the importance of various inputs, which relaxes the search space to be continuous. Moreover, architecture weights are introduced into CLNN to make the formula structure search differentiable. wCL formula-informed intensity rates are carefully designed so that the parameters appearing in the rules can be learned through maximum likelihood estimation using gradient-based approaches. CLNN is tested on synthetic datasets to show that CLNN can recover the ground-truth rules as well as on real-world datasets to demonstrate its model-fitting performance.

## 2 PRELIMINARIES

### 2.1 NOTATION & BACKGROUND

Let $\mathcal{L}$ denote the set of event labels, and $M = |\mathcal{L}|$ denote the number of event labels. An event stream is a sequence of events including time stamps, denoted as $\mathcal{D} = \{(l_1, t_1), (l_2, t_2), ..., (l_N, t_N)\}$, where $t_i \in \mathbb{R}^+$ denotes a time stamp between the beginning time $t_0 = 0$ and end time $t_{N+1} = T$, and $l_i \in \mathcal{L}$ is the event label that happens at $t_i$. We refer to 'event label' and 'label' interchangeably. Every event label $l \in \mathcal{L}$ has an associated conditional intensity rate describing the occurrence rate of label $l$ at $t$ given the history up to $t$. In multivariate temporal point processes, conditional intensity rates describe the dynamics of events. Let $\mathcal{H}_t = \{(l_i, t_i) : t_i < t\}$ denote the historical events up to time $t$. The conditional intensity rate of event label $l$ is denoted as $\lambda_l(t|\mathcal{H}_t)$. Specifically, $\lambda_l(t|\mathcal{H}_t)$ describes the expected number of occurrences of event label $l$ in an infinitesimal interval $[t, t+\Delta t]$ given the history $\mathcal{H}_t$, i.e., $\lambda_l(t|\mathcal{H}_t) = \lim_{\Delta t \to 0}(E[N_l(t + \Delta t) - N_l(t)|\mathcal{H}_t]/\Delta t)$, where $N_l(t)$ denotes the number of event label $l$'s occurrences up to $t$.

**Example 1** *A running example of an event stream with* 11 *events of* 4 *labels is shown in Figure 1(a).*

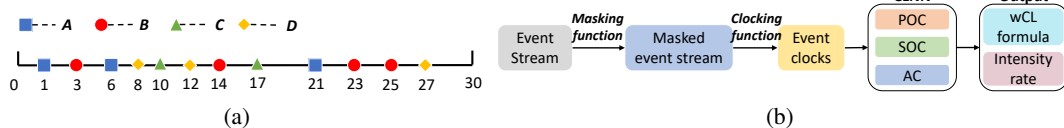

|     |     |
|:---:|:---:|
| (a) | (b) |

Figure 1: ($a$): An event stream example with $N = 11$ events of $M = 4$ event labels over $T = 30$ days. (Integer-valued time stamps are utilized for easy interpretation, note that the proposed approach also works for $t_i \in \mathbb{R}$). ($b$): The overall workflow of the proposed method (POC: paired order cell, SOC: singleton order cell, AC: architecture cell, details presented in Section 2.2 to 3.3).

### 2.2 ORDER REPRESENTATIONS FOR EVENT STREAMS

The overall workflow of the proposed framework is visualized as Figure 1(b). The raw event streams first go through a masking function to generate the masked event streams, which are then transformed into event clocks using a clocking function. The event clocks are given as inputs to the clock logic neural network (CLNN) to learn interpretable wCL formulas and the intensity rate of event occurrences. The following sections provide a detailed explanation for each module in Figure 1(b).

We are interested in exploring the effect of temporal ordering between event labels and the occurrences of causal event labels in a historical window on the occurrence rate of a particular event label,

where the generative mechanism is expressed as interpretable formulas. An event stream up to $t$ may include multiple occurrences of the same event label, thus a masking function is required to mask out duplicated event labels in the history for accessing the ordering information at any $t$. Here we adopt a technique similar to Bhattacharjya et al. (2020) for extracting distinct event labels from $\mathcal{H}_t$.

**Definition 1 (Masking Function)** *A masking function $\Gamma(\cdot)$ is a function that takes an event stream as input and returns a new event stream that is a subset of the input stream and contains no duplicated event labels. Mathematically, $\Gamma(\cdot)$ is applied to $\mathcal{H}_t = \{(l_i, t_i)\}$ and converts it into a new stream $\mathcal{H}'_t = \{(l_j, t_j) \in \mathcal{H}_t : l_j \neq l_{j'} \text{ if } j \neq j'\}$.*

We consider the following two masking functions as per Bhattacharjya et al. (2020) due to simplicity: 'first' masking and 'last' masking. The 'first' (resp. 'last') masking function keeps the first (resp. last) occurrence of an event label in an event stream.

**Example 1 (cont.)** *Let $\mathcal{H}_{13} = \{(A, 1), (B, 3), (A, 6), (D, 8), (C, 10), (D, 12)\}$. The 'first' masking function converts it to $\mathcal{H}'_{13} = \{(A, 1), (B, 3), (D, 8), (C, 10)\}$, and the 'last' masking function converts it to $\mathcal{H}'_{13} = \{(B, 3), (A, 6), (C, 10), (D, 12)\}$.*

With the masked event history $\mathcal{H}'_t$, we define two order representations for the order relationship between any two event labels and the occurrence of an event within a historical window of $t$.

**Definition 2 (Paired Order Representation (POR))** *A paired order representation is defined as $[l_i, l_j] \in [\mathcal{L}]^2$, where $[\mathcal{L}]^2$ denotes two-element permutation of a subset of $\mathcal{L}$. A paired order representation for $\mathcal{H}'_t$ can be obtained by arranging any two distinct labels in $\mathcal{H}'_t$ in a sequential order.*

**Definition 3 (Singleton Order Representation (SOR))** *A singleton order representation is denoted as $[l_j, \underline{u}_{l_j}] \in \mathcal{L} \times \mathbb{R}_+$, representing event label $l_j \in \mathcal{L}$ occurred within the past $\underline{u}_{l_j}$ time units, where $\underline{u}_{l_j}$ is a variable to learn through a process that will be explained in Section 3.3.*

**Example 1 (cont.)** *With first masking, an example of paired order representation for $\mathcal{H}'_{13}$ can be $[A, B]$ representing "A happens before B" or $[B, C]$ representing "B happens before C". The overall order representation for $\mathcal{H}'_{13}$ is expressed as $[A, B, D, C]$, which can be derived from the paired order representations: $[A, B], [B, D], [D, C]$. A singleton order representation example of $\mathcal{H}'_{13}$ can be expressed as $[B, 10.5]$, meaning B happened in the past $10.5$ days.*

## 2.3 Weighted Clock Logic Formula

To adapt $\mathcal{H}'_t$ to continuous-time signals that can be described by logical statements, we extract clock signals from $\mathcal{H}'_t$ to describe the time passed since the last occurrence of a label. A clocking function is introduced to convert $t_j$ into a clock signal $c_j$ denoting the time interval length between $t_j$ and $t$.

**Definition 4 (Clocking Function)** *A clocking function $\Xi(\cdot)$ converts $\mathcal{H}'_t$ into a vector of clock signals as $\mathcal{C}'(t) = [c_1(t), c_2(t), ..., c_M(t)]^T \in \mathbb{R}_+^M$ with $c_i(t)$ denoting the clock signal for event label $i \in \mathcal{L}$, where $c_i(t)$ is computed as $c_i(t) = t - t_j$ if $(l_j, t_j) \in \mathcal{H}'_t$ and $l_j = i$, and $c_i(t) = \bar{Z}$ otherwise. Note that $\bar{Z}$ is a user-defined, large positive number to indicate event label $i$ not happening in $\mathcal{H}'_t$.*

**Example 1 (cont.)** *Taking the 'first' masked event stream $\mathcal{H}'_{13} = \{(A, 1), (B, 3), (D, 8), (C, 12)\}$ as an example, the event clocks are extracted as $\mathcal{C}'(13) = [12, 10, 1, 5]^T$.*

The event clocks can essentially provide the ordering between any two event labels in that the difference between any two event labels' clock signals reflects which event label happens first. As shown in the diabetes prediction example in the Introduction section, the time interval between ordering events is notably important in explaining and predicting an event label's occurrence. In contrast to (Li et al., 2020; 2021) which only learns the temporal ordering relation between event labels, we define a paired order predicate (POP) with a learnable parameter $\underline{u}_{l_i l_j}$ to describe the time interval between two ordered event labels $l_i$ and $l_j$ and a singleton order predicate (SOP) with a learnable parameter $\underline{u}_{l_j}$ to describe the occurrence of label $l_j$ within a historical window $\underline{u}_{l_j}$ as follows.

**Definition 5 (Paired Order Predicate)** *A POP describes the order between two labels $l_i, l_j \in \mathcal{L}, l_i \neq l_j$, denoted as $\pi_{pop}^{l_i l_j} := g(c_{l_i}, c_{l_j}) = c_{l_i} - c_{l_j} > \underline{u}_{l_i l_j}$, where $\underline{u}_{l_i l_j} \in \mathbb{R}$ is a parameter to learn. A positive $\underline{u}_{l_i l_j}$ means $l_i$ happened before $l_j$ for at least $\underline{u}_{l_i l_j}$ time units, and a negative $\underline{u}_{l_i l_j}$ means $l_j$ happened before $l_i$ for at most $-\underline{u}_{l_i l_j}$ time units. A POP is used in the POC of Figure 1(b).*

**Definition 6 (Singleton Order Predicate)** *An SOP describes a causal label $l_j \in \mathcal{L}$ occurring within the past $\underline{u}_{l_j}$ time units, defined as $\pi_{sop}^{l_j} := c_{l_j} - \underline{u}_{l_j} < 0$, where $\underline{u}_{l_j} \in \mathbb{R}_+$ is a learnable parameter.*

Instead of taking a heuristic approach for some underlying combinatorial search problem for a given set of temporal predicates (Bhattacharjya et al., 2020; 2021; Li et al., 2021) to uncover the effective order relations, this work proposes a differentiable learning model to learn suitable singleton and paired order predicates among all the possible choices of order predicates through a gradient-based approach. The scheme of weighted signal temporal logic (wSTL) in Yan et al. (2021; 2022) is exploited to build weighted clock logic (wCL) formulas that are logical compositions of singleton and paired order predicates. The syntax of wCL is recursively defined as (Mehdipour et al., 2021):

$$\phi := \pi_{pop}^{l_i l_j} \,|\, \pi_{sop}^{l_j} \,|\, \neg\phi \,|\, \phi_1^{w_1} \wedge \phi_2^{w_2} \cdots \wedge \phi_k^{w_k} \,|\, \phi_1^{w_1} \vee \phi_2^{w_2} \cdots \vee \phi_k^{w_k}, \tag{1}$$

where $\phi_1, \cdots \phi_k$ are wCL formulas, $\neg$ denotes negation, $\wedge$ denotes logical conjunction, $\vee$ denotes logical disjunction, $w_j \geq 0, j = 1, \cdots, k$ denotes non-negative weights assigned to $\phi_1, \cdots, \phi_k$ in the conjunction and disjunction operations. A wCL formula can describe the characteristics of $\mathcal{H}_t$, thus the conditional intensity rate of event $l$ given $\mathcal{H}_t$ can be equivalently denoted as $\lambda_{l|\phi}(t)$.

**Remark 7** *The syntax above means each wCL formula can be built by using predicates in $\pi_{pop}^{l_i l_j}$ or $\pi_{sop}^{l_j}$ and then by recursively applying the $\neg$ or the $\wedge$ or the $\vee$ operations.*

**Example 1 (cont.)** *A wCL formula example is $\phi = (c_A - c_B > 1)^1 \wedge (c_C < 3)^{0.05}$. The first and second clauses read "A happened before B for at least one day" and "C happened less than 3 days ago", respectively. Note that $\phi$ is satisfied by the event stream up to $t = 13$ in Figure 1(a). The two clauses have weights of $1$ and $0.05$, reflecting the first clause is more important than the second one.*

## 3 WEIGHTED CLOCK LOGIC POINT PROCESSES

### 3.1 TRUTH DEGREE OF WEIGHTED CLOCK LOGIC

To quantitatively measure the satisfaction degree of a wCL formula $\phi$ over the event clocks $\mathcal{C}'(t)$, i.e., how well does $\phi$ describe the underlying patterns of $\mathcal{C}'(t)$, we propose smooth activation functions (AFs) to compute the truth degree, denoted $p(\mathcal{C}', \phi, t) \in [0, 1]$, defined as (Riegel et al., 2020):

$$p(\mathcal{C}', \pi_{pop}^{l_i l_j}, t) = sigmoid(c_{l_i}(t) - c_{l_j}(t) - \underline{u}_{l_i l_j}), \tag{2}$$

$$p(\mathcal{C}', \pi_{sop}^{l_j}, t) = sigmoid(\underline{u}_{l_j} - c_{l_j}(t)), \tag{3}$$

$$p(\mathcal{C}', \neg\phi, t) = 1 - p(\mathcal{C}', \phi, t). \tag{4}$$

In contrast to the combinatorial search of the temporal logic predicates in Li et al. (2021), the smooth design of AFs in (2) - (4) benefits the maximum likelihood estimation problem shown later in Section 3.6 by allowing it to learn the parameters in the POP and SOP through gradient-based methods. Next, we present the design of activation functions (AF) for the $\wedge$ operator. Here we use a 2-ary conjunction operator to motivate the design. Let $p^\wedge = p(\mathcal{C}', \phi_1^{w_1} \wedge \phi_2^{w_2}, t) \in [0, 1]$. Intuitively, $p^\wedge$ is *low* when either input is *low*, and $p^\wedge$ is *high* when both inputs are *high*. Here we adopt a similar idea to Sen et al. (2022) for capturing the *low* and *high*. A user-defined hyperparameter $\alpha \in [\frac{1}{2}, 1]$ is introduced to aid the interpretability of *low* and *high* such that $p^\wedge$ represents *high* if $p^\wedge \in [\alpha, 1]$ and *low* if $p^\wedge \in [0, 1 - \alpha]$. Considering the importance weights, a *low* input with a zero weight should not impact the output, which implies $p^\wedge$ should be *low* when both inputs are *low*. With these considerations, the AF for the $\wedge$ operator is defined as follows: (See Appendix A for more details.)

$$p(\mathcal{C}', \phi_1^{w_1} \wedge \phi_2^{w_2} \cdots \wedge \phi_k^{w_k}, t) \quad = f\left(\beta - \sum_{j=1}^{k} w_j(1 - p(\mathcal{C}', \phi_j, t))\right), \tag{5}$$

$$\text{subject to} \quad \beta - \sum_{j=1}^{k} w_j(1 - \alpha) \geq \alpha, \beta - \sum_{j=1}^{k} w_j \alpha \leq 1 - \alpha,$$

where $f(z) = \max\{0, \min\{z, 1\}\}$ clamps the truth degree into $[0, 1]$, $w_j \geq 0$ and $\beta \geq 0$ are parameters to learn. By De Morgan's law (Hurley, 2014), the AF for the $\vee$ operator is defined as

$$p(\mathcal{C}', \phi_1^{w_1} \vee \phi_2^{w_2} \cdots \vee \phi_k^{w_k}, t) \quad = f\left(1 - \beta + \sum_{j=1}^{k} w_j(p(\mathcal{C}', \phi_j, t))\right), \tag{6}$$

$$\text{subject to} \quad 1 - \beta + \sum_{j=1}^{k} w_j \alpha \geq \alpha, 1 - \beta + \sum_{j=1}^{k} w_j(1 - \alpha) \leq 1 - \alpha.$$

An event stream with $M$ event labels would generate $\mathrm{P}_M^2 = \frac{M!}{(M-2)!}$ paired order predicates and $M$ singleton order predicates. If a conjunction or disjunction operator takes these predicates as inputs, how it recognizes the effective order predicates in describing the event dynamics becomes a critical issue. By carefully designing the AFs in (5) - (6), the logical operators exhibit the following properties so as to recognize effective inputs. This is a critical advantage over Bhattacharjya et al. (2020; 2021); Li et al. (2021) in that it allows a differentiable search of the suitable predicates among all the possible choices of order predicates in an end-to-end manner. Here we illustrate the properties for $\wedge$ with two inputs, which can be generalized to $k$-ary inputs. (See Appendix B for more details.)

**Theorem 8** *The AF for the $\wedge$ operator with two inputs exhibits the following properties.*

*1) Nonimpact for zero weights: If $w_j = 0, j = 1, 2$, $p(\mathcal{C}', \phi_j, t)$ has no impact on $p(\mathcal{C}', \phi_1 \wedge \phi_2, t)$.*

*2) Impact ordering: If $p(\mathcal{C}', \phi_1, t) = p(\mathcal{C}', \phi_2, t)$, and $w_1 \geq w_2$, then $\frac{\partial p(\mathcal{C}', \phi_1 \wedge \phi_2, t)}{\partial p(\mathcal{C}', \phi_1, t)} \geq \frac{\partial p(\mathcal{C}', \phi_1 \wedge \phi_2, t)}{\partial p(\mathcal{C}', \phi_2, t)}$.*

*3) Monotonicity: $f(\beta - \sum_{j=1}^2 w_j(1 - p(\mathcal{C}', \phi_j, t))) \leq f(\beta - \sum_{j=1}^2 w_j(1 - (p(\mathcal{C}', \phi_j, t) + d)))$, $d \geq 0$.*

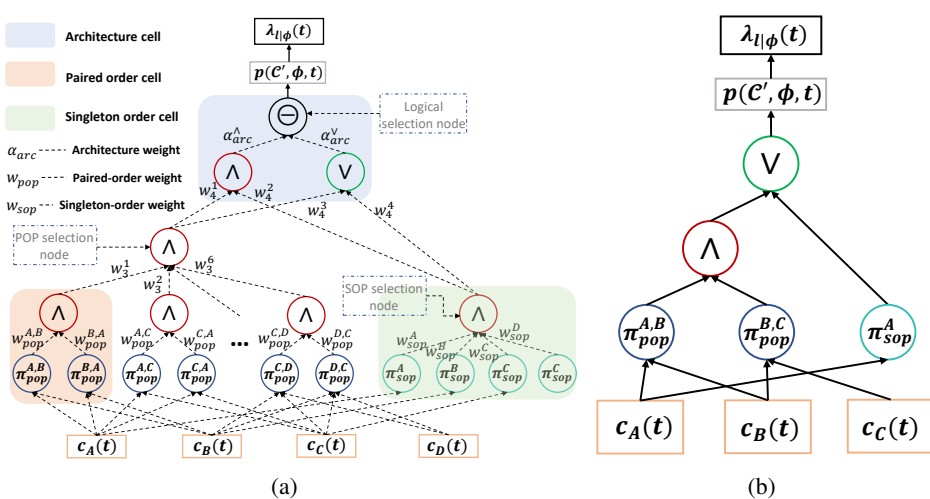

Figure 2: CLNN Structure. $(a)$: Continuous relaxation of the search space using weights. $(b)$: The learned discrete model structure for $\phi = (\pi_{pop}^{A,B} \wedge \pi_{pop}^{B,C}) \vee (\pi_{sop}^A)$.

## 3.2 LEARNING OF PAIRED ORDER REPRESENTATION

With the smooth AFs designed in (2) - (6), a neuro-symbolic model called clock logic neural network (CLNN) can be designed for any given wCL formula $\phi$, in which every neuron has a corresponding symbolic representation. A typical CLNN for $\phi = (\pi_{pop}^{A,B} \wedge \pi_{pop}^{B,C}) \vee (\pi_{sop}^A)$ is visualized as Fig. 2(b), which can be considered as the discrete structure obtained by learning the parameters of the model in Figure 2(a) and keeping the dominant components. Here $\phi$ can be interpreted as "($A$ happens before $B$ for at least $\underline{u}_{AB}$ time units or $B$ happens before $C$ for at least $\underline{u}_{BC}$ time units) and $A$ happens within the past $\underline{u}_A$ time units." This part describes the continuous relaxation of the search space by designing a paired order cell, a singleton order cell, and an architecture cell for learning the paired order representation, singleton order representation and the formula structure.

**Paired Order Cell (POC).** A POC is a directed acyclic graph (DAG) comprising two paired order predicate (POP) nodes and one logical node for the $\wedge$ operator, shown as an orange block in Figure 2(a). The two POP nodes represent $\pi_{pop}^{l_i,l_j}$ and $\pi_{pop}^{l_j,l_i}$ sharing the same parameter $\underline{u}_{l_i,l_j}$, where $\pi_{pop}^{l_i,l_j}$ denotes "$l_i$ happened before $l_j$ for at least $\underline{u}_{l_i,l_j}$ time units" and $\pi_{pop}^{l_j,l_i}$ denotes "$l_j$ happened before $l_i$ for at least $\underline{u}_{l_i,l_j}$ time units". Each POP has an associated weight $w_{pop}^{l_i,l_j}$ or $w_{pop}^{l_j,l_i}$ to be learned, and the $\wedge$ operator forces one of the two weight parameters to dominate the other one such that the learned POR is consistent with the event stream. For example, the POC in Figure 2(a) aims to learn the POR between $A$ and $B$, whose discretized version would be either $\pi_{pop}^{A,B}$ or $\pi_{pop}^{B,A}$. An event stream with $M$ event labels can generate $\mathrm{P}_M^2 = \frac{M!}{(M-2)!}$ PORs between any two event

labels, resulting in $(\mathrm{P}_M^2/2)$ POCs. Similar to learning the POR between any two events, the discrete order representations for the entire history $\mathcal{H}_t$ can be learned using a POP selection node (as shown in Figure 2(a)) that takes the outputs of all the POCs as input and identifies the important PORs. The learning of the POCs essentially becomes learning the $w$, $\beta$ in (5) for the POCs and the POP selection node, as well as $\underline{u}_{l_i l_j}$ in (2) for the POPs through back propagation. The discrete PORs can be acquired by keeping the top-$k$ strongest POCs and the dominant POPs.

### 3.3 LEARNING OF SINGLETON ORDER REPRESENTATION

**Singleton Order Cell (SOC).** The learning of SOR is accomplished by an SOC, which is displayed as a green block in Figure 2(a). An SOC is a DAG comprising $M$ singleton order predicate (SOP) nodes and one SOP selection node for the $\wedge$ operator. An SOP node represents $\pi_{sop}^{l_j}$ that takes $c_{l_j}(t)$ as input and returns the truth degree of $\pi_{sop}^{l_j}$ over $c_{l_j}(t)$. The SOP selection node has the same functionality as the POP selection node. The $\wedge$ operator in the SOP selection node assigns a nonnegative weight to every SOP node and learns the importance weights $w$ and $\beta$ to extract the dominant SORs affecting the conditional intensity rate the most. The learning of the SOC is thus learning the $w, \beta$ in (5) for the SOP selection node and $\underline{u}_{l_j}$ in (3) for the SOPs through back propagation. The discrete SORs can be determined by keeping the top-$k$ strongest SOPs.

### 3.4 LEARNING OF FORMULA STRUCTURE

**Architecture Cell (AC).** For a given set of PORs or SORs, their conjunction or disjunction will behave differently and have distinct meanings. For instance, given two causal formulas $\phi_1 = (c_A - c_B > 1)^1 \wedge (c_C < 5)^1$ and $\phi_2 = (c_A - c_B > 1)^1 \vee (c_C < 5)^1$ for the occurrence of event label $D$, $\phi_1$ means "($A$ happens before $B$ for at least 1 time unit) **and** ($C$ happens within the past 5 time units) simultaneously will cause $D$ to happen", whereas $\phi_2$ means "($A$ happens before $B$ for at least 1 time unit) **or** ($C$ happens within the past 5 time units) alternatively will cause $D$ to happen." The afore-mentioned cells can learn the order representations. Nevertheless, whether their outputs should be connected by the $\wedge$ or $\vee$ operator needs to be determined. Here we consider the outputs of the POCs and the SOCs having two choices of being connected by a $\wedge$ or $\vee$ operator, each of which is associated with an architecture weight $\alpha_{arc}^\wedge$ or $\alpha_{arc}^\vee$ that enables continuous learning of the two choices; this is also called differentiable architecture search (Liu et al., 2019). An architecture cell is introduced for learning the model architecture, which comprises two logical nodes representing a $\wedge$ operator and a $\vee$ operator as well as a logical selection node (LSN), shown as the blue block in Figure 2(a). Let $\boldsymbol{p} = \{p_1, ..., p_k\}$ denote the set of inputs for each logical operator. Subsequently, the conjunction operator takes $\boldsymbol{p}$ as input and returns $p^\wedge = f(\beta^\wedge - \sum_{j=1}^{k} w_j^\wedge(1 - p_j))$, and the disjunction operator takes $\boldsymbol{p}$ as input and returns $p^\vee = f(1 - \beta^\vee + \sum_{j=1}^{k} w_j^\vee p_j)$. The LSN represented by $\ominus$ takes $p^\wedge$ and $p^\vee$ as inputs and returns their weighted sum, where the weights are computed using the softmax of the architecture weights as shown below:

$$p_\ominus = p(\mathcal{C}', \phi, t) = \sum_{m \in \{\wedge, \vee\}} \frac{e^{\alpha_{arc}^m}}{\sum_{m' \in \{\wedge, \vee\}} e^{\alpha_{arc}^{m'}}} p^m. \tag{7}$$

The task of architecture search then reduces to learning the architecture weights $\alpha_{arc}^\wedge$, $\alpha_{arc}^\vee$ and the $w, \beta$ in (5) - (6) for the two logical operators, which can be executed simultaneously while learning parameters in the POCs and SOCs. The outcome of the architecture search process is a discrete architecture obtained by retaining the logical operator with the strongest architecture weight.

### 3.5 wCL-INFORMED INTENSITY FUNCTION

The output of a CLNN is the truth degree of $\phi$ over $\mathcal{C}'$ at $t$, which is incorporated into modeling the conditional intensity rates. The modeling process aims to discover the generative mechanism as wCL formulas for every $l \in \mathcal{L}$. In other words, a larger value of $p(\mathcal{C}', \phi, t)$ should reflect that $\phi$ has a greater impact on the occurrence of a particular label. For example, if the wCL formula for affecting the occurrence of event label $D$ is given as $\phi = ((\pi_{pop}^{A,B})^{w_1} \wedge (\pi_{sop}^C)^{w_2})$, it means if $\phi$ is satisfied or the truth degree of $\phi$ is *high*, then it has a strong impact on the occurrence of $D$, where the impact can be promoting or inhibiting the occurrence of $D$. In terms of the relation between the truth degree and the conditional intensity rate, the higher the truth degree $p(\mathcal{C}', \phi, t)$, the greater its impact on $\lambda_{D|\phi}$. Note

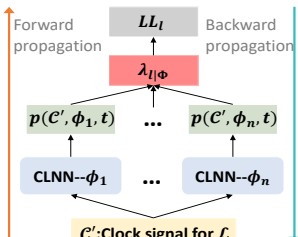

Figure 3: The overall learning framework for $n$ wCL formulas.

that the occurrence of one event label may depend on multiple wCL formulas. This work follows the assumption that the impact of multiple formulas are additive in predicting the intensity rate, similar to Li et al. (2020). To incorporate a set of wCL formulas $\Phi = \{\phi_1, \phi_2, ..., \phi_n\}$ into the modeling of the conditional intensity rate, we define a wCL formula-informed conditional intensity rate as:

$$\lambda_{l|\Phi}(t) = \exp(\sum_{i=1}^{n} w_{\phi_i} p(\mathcal{C}', \phi_i, t) + \rho), \tag{8}$$

where $w_{\phi_i}$ is the weight of $\phi_i$, and $\rho$ is a bias term that allows for spontaneous occurrence without the influence from $\phi$.

## 3.6 MAXIMUM LIKELIHOOD ESTIMATION

Suppose event stream $\mathcal{D}$ contains $n_l$ occurrences of event $l$, for which the occurrence time stamps are denoted as $t_{l_1}, t_{l_2}, ..., t_{l_{n_l}}$. Let $t_0 = 0, t_{l_{n_l+1}} = T$. Based on the conditional intensity function in (8), the likelihood for label $l$ over the event stream is calculated as (Daley & Vere-Jones, 2003):

$$L_l = \prod_{i=0}^{n_l-1} \left( \exp\left( -\int_{t_{l_i}}^{t_{l_{i+1}}} \lambda_{l|\Phi}(s)ds \right) \lambda_{l|\Phi}(t_{l_{i+1}}) \right) \exp\left( -\int_{t_{l_{n_l}}}^{T} \lambda_{l|\Phi}(s)ds \right). \tag{9}$$

The corresponding log-likelihood for event label $l$ is expressed as $LL_l = (-\int_0^T \lambda_{l|\Phi}(s)ds) + \sum_{i=1}^{n_l}[\log(\lambda_{l|\Phi}(t_{l_i}))]$. The total log-likelihood of all the events in $\mathcal{D}$ is thus $LL_{\mathcal{D}} = \sum_{l \in \mathcal{L}} LL_l$. During the training process, we train the model parameters for each event label separately. Specifically, the maximum likelihood estimation problem for event label $l$ can be formulated as follows:

$$\min \quad -LL_l \tag{10}$$

$$s.t. \quad \forall \phi \in \Phi, \forall 1 \le k \le K_\phi^\wedge, \beta_k - \sum_{i \in I_k} w_{i,k}(1-\alpha) \ge \alpha, \beta_k - \sum_{i \in I_k} w_{i,k}\alpha \le 1 - \alpha, \tag{11}$$

$$\forall \phi \in \Phi, \forall 1 \le k' \le K_\phi^\vee, 1 - \beta_{k'} + \sum_{i \in I_{k'}} w_{i,k'}\alpha \ge \alpha, 1 - \beta_{k'} + \sum_{i \in I_{k'}} w_{i,k'}(1-\alpha) \le 1 - \alpha, \tag{12}$$

$$w_{i,k} \ge 0, \beta_k \ge 0, w_{i,k'} \ge 0, \beta_{k'} \ge 0, \underline{u}_{l_j} \ge 0,$$

where $K_\phi^\wedge$ (resp. $K_\phi^\vee$) is the number of $\wedge$ (resp. $\vee$) operators in $\phi$, $I_k$ (resp. $I_{k'}$) denotes the inputs to the $k$-th $\wedge$ (resp. $k'$-th $\vee$) operator. Please see Appendix A for more details about the above formulation. The overall learning framework is shown in Figure 3, in which the forward propagation computes $LL_l$ by using $n$ CLNNs; each learns a wCL formula $\phi_i$ and the backward propagation updates the parameters in $n$ CLNNs using projected gradient descent.

## 4 EXPERIMENTS

We conduct several experiments on synthetic and real-world datasets to demonstrate the efficacy of our proposed model. Simultaneously, we compare with state-of-the-art (SOTA) models. The experiments are run using the AdamW optimizer in Pytorch (1.10.2) on a Windows 10 system desktop with a 16-core CPU (i7, 3.60GHz) and 32 GB RAM. Our code is available at https://ICLR-CLNN.

## 4.1 MODELS

**Multivariate Hawkes Process (MHP)** [(Bacry et al., 2017)]: A conventional multivariate Hawkes process utilizing an exponential kernel function to describe the conditional intensity rate, which involves a decay rate and an infectivity matrix characterizing the inter-dependence among events. This model is implemented in the *tick*[1] library, where the learning problem is posed as a convex quadratic programming problem with a fixed decay rate.

**Proximal Graphical Event Model (PGEM)** [(Bhattacharjya et al., 2018)]: A type of GEM that models event data by considering whether a parent in some underlying graph happens in a proximal (recent) window.

---

[1]https://x-datainitiative.github.io/tick/modules/hawkes.html

| | |
|---|---|
| Ground truth | $\hat{\phi}_1 = (c_A - c_B > 1)^1 \wedge (c_A - c_C > 3)^1$ |
| CLNN's rule | $(\mathbf{c_A - c_B > 1.21})^{\mathbf{1.52}} \wedge (\mathbf{c_A - c_C > 3.00})^{\mathbf{1.41}} \wedge (c_A - c_D > 0.82)^{0.33} \wedge (c_B - c_C > 4.33)^0 \wedge (c_B - c_D > 10.69)^0 \wedge (c_D - c_C > -6.57)^{0.16}$ |
| TELLER's rule | $A$ before $D$, $B$ before $D$, $C$ before $D$, $A$ before $D$ and $C$ before $D$ |
| OGEM-tab's rule | *Excitation*: $[B]$, $[C, B]$, $[B, C]$; *Inhibitory*: $[A]$, $[C, A]$, $[A, C]$ |

Table 1: Comparison of rule discovery for CLNN, TELLER, and OGEM-tab on the Syn-1 dataset.

**Ordinal Graphical Event Model (OGEM)** [(Bhattacharjya et al., 2020; 2021)]: An ordinal GEM that models the impact of the order of events on the conditional intensity rate. OGEM-tab (resp. OGEM-tree) refers to an OGEM that adopts a tabular (resp. tree) representation of orders.
**Temporal Logic Rule Learner (TELLER)**[2] [(Li et al., 2021)]. This is a method to learn first-order temporal logic rules explaining the generative mechanism of TPPs. The rule discovery process is formulated as a maximum likelihood estimation problem solved by a branch-and-price algorithm.

## 4.2 Synthetic Datasets

The first part of this experiment demonstrates CLNN's capability of recovering ground-truth rules using three synthetic datasets generated by CLNN with pre-specified formula structure and parameters, including $\underline{u}_{l_i l_j}$ in $\pi_{pop}^{l_i, l_j}$, as well as the importance weights $w$ and bias $\beta$ in (5) for logical operators, and the $w_\phi$ and $\rho$ in (8) for the conditional intensity rate.

**Experimental Setting.** Each synthetic dataset contains $1,000$ event streams partitioned into three sets: training (70%), validation (15%), and test (15%). Every dataset is generated using a wCL formula with $w_\phi = 3$ and $\rho = -5$. The truth value threshold is set as $\alpha = 0.5$, and the clock signal for representing an event not occurring in $\mathcal{H}'_t$ is set as $\bar{Z} = 1.5T_{\max}$, where $T_{\max}$ is the maximal ending time among all the event streams. During the training process, we initialize the parameters using four approaches (see Appendix C.5 for more details) and report the best one, and CLNN aims to recover the manually set parameters.

**Results.** The ground-truth rule $\hat{\phi}_1$ for generating the first synthetic dataset (Syn-1) with $\mathcal{L} = \{A, B, C, D\}$ and the rules discovered by CLNN, TELLER, and OGEM-tab are summarized in Table 1. Results for the other synthetic datasets are presented in Appendix C. The rules are learned using the 'last' masking method, which was also used for data generation. The experimental results show an accurate recovery performance of CLNN in terms of order representation recovery and parameter identification. The unweighted version of the ground truth rule reads: "If $A$ happens before $B$ for at least 1 time unit and $A$ happens before $C$ for at least 3 time units, then $D$ will happen". The rule of TELLER only reflects the temporal relation between events $A, B, C$ and $D$ but is unable to capture the temporal relation between $A$ and $B$ or $A$ and $C$, which does not match the ground-truth rule. In OGEM-tab's rule, $[l]$ denotes a single parent. We show the top 3 excitation and inhibitory rules from OGEM-tab, where excitation (resp. inhibitory) means $\lambda_{l|\Phi}$ is higher (resp. lower) than the $\lambda_{l|\Phi}$ with all $w_{\phi_i} = 0$. The excitation rules of OGEM-tab do not match the ground-truth rule. In contrast, the rule discovered by CLNN ($\phi_1$) assigns larger weights to the paired order predicates $\pi_{pop}^{A,B} = (c_A - c_B > 1.21)$ and $\pi_{pop}^{A,C} = (c_A - c_C > 3.00)$ and small weights to the other predicates, where the interval values of 1.21 and 3.00 are both learned. By ignoring the small weights, $\phi_1$ can be interpreted as "If $A$ happens before $B$ for at least 1.21 time units and $A$ happens before $C$ for at least 3.00 time units, then $D$ will happen", meaning the paired order representations discovered by CLNN match well with the ground truth. Moreover, CLNN's rules are more expressive than TELLER and OGEM as it provides a detailed interval length between two ordered labels.

To show the computational efficiency of our gradient-based learning, we compare the runtimes of CLNN and TELLER on the synthetic datasets in Table 2. Notably, CLNN not only recovers the correct order representations but also was two orders of magnitude faster on average (5.62 s vs 635.99 s). In addition, CLNN can learn more expressive order representations that describe both the order relation between two events and their interval length.

| wCL formula | $\phi_1$ | $\phi_2$ | $\phi_{3,1}$ | $\phi_{3,2}$ | Average |
|---|---|---|---|---|---|
| CLNN | 5.20 | 4.60 | 4.95 | 7.73 | 5.62 |
| TELLER | 252.91 | 286.83 | 925.58 | 1078.66 | 635.99 |

Table 2: Runtime (s) for CLNN and TELLER on synthetic datasets.

## 4.3 Real-world Datasets

**LinkedIn** [(Xu et al., 2017)]. An event dataset related to job hopping records of $3,000$ LinkedIn users in 82 IT companies. Each event stream records a user's check-in time stamps for different companies or the time stamps for role change within the same company. We filter the dataset to popular companies as per Bhattacharjya et al. (2020), resulting in $1,000$ users.

---

[2]https://github.com/FengMingquan-sjtu/Logic_Point_Processes_ICLR

**Mimic II** [(Saeed et al., 2011)]. An event dataset concerning health records of patients from Intensive Care Unit (ICU) visits over 7 years. A patient's event stream records each visit's time stamp and the corresponding diagnosis. We filter out sequences with few visits, resulting in 650 patients.

**Stack Overflow** [(Grant & Betts, 2013)]. An event dataset that is related to the badges awarded to users in the question-answering website, the Stack Overflow. Each user's event stream records the badges that he/she receives at various time stamps. We keep the event streams with one or more of 20 types of badges and sample $1,000$ users from the dataset used in Du et al. (2016).

**Experimental Setup.** Each dataset is partitioned into three sets: training (70%), validation (15%), and test (15%). For simplicity, $\underline{u}_{l_i l_j}$ are set as $0$ to study the ordering representations. The truth value threshold is $\alpha = 0.5$, and $\bar{Z} = 1.5T_{\max}$, same as the setting for the synthetic datasets, and the number of subformulas is $n = 5$, and the parameters are initialized as random numbers from a uniform distribution on $[0, 1)$. CLNN is trained on the training set, and the validation set is utilized for model selection during training. Model fit is evaluated using log-likelihood on the test set.

**Results.** We follow a similar trend to Bhattacharjya et al. (2018; 2020; 2021) to use the log-likelihood for evaluation of the model's performance. The log-likelihood on the real-world datasets is reported in Table 3, where *DR* denotes the difference ratio – the difference between CLNN and the best SOTA divided by the absolute value of best SOTA. CLNN's result is chosen as the better one among the 'first' or the 'last' masking. Notably, CLNN outperforms the baseline models on the LinkedIn dataset (13.40% advantage) and achieves a competitive result on the MIMIC II dataset (1.63% loss only). It is observed that PGEM achieves a better result on the Stack Overflow dataset. In Stack Overflow, one type of badge can be awarded only when a user receives a particular badge multiple times, for example, the 'Epic' badge is awarded only when earning 200 daily reputations 50 times, depending on the 'Mortarboard' badge acquired while answering or asking questions. CLNN and OGEMs apply masking methods to the data, which may not capture the above dependence. In contrast, PGEM models data without masking, making it more suitable for this dataset.

| Dataset | $N$ (# events) | $M$ (labels) | MHP | PGEM | OGEM-tab | OGEM-tree | TELLER | CLNN | *DR* |
|---|---|---|---|---|---|---|---|---|---|
| LinkedIn | 2932 | 10 | -1593 | -1462 | -1478 | -1418 | -1548 | **-1228** | *13.40%* |
| MIMIC II | 2419 | 15 | -567 | -500 | -474 | **-429** | -645 | -436 | *-1.63%* |
| Stack Overflow | 71254 | 20 | -52543 | **-48323** | -49344 | -49192 | -71101 | -50981 | *-5.50%* |

Table 3: Dataset information and log-likelihood for all models on the real-world datasets.

**Case Study.** The primary strength of CLNN over the SOTA models is that it can describe the generative mechanism as wCL formulas, being more expressive and potentially providing more detailed information. CLNN can be deployed as a valuable tool for assisting domain specialists in knowledge discovery from event data. Here we showcase the above strength of CLNN using an il-

| | Rules | Effect |
|---|---|---|
| CLNN | $\phi_1 = (c_D > c_H)^{0.90} \wedge (c_I > c_J)^{0.72}$ | Inhibitory |
| | $\phi_2 = ((c_B < 0.45)^{0.58} \wedge (c_D < 0.05)^{0.66}$ | Excitation |
| | $\phi_3 = (c_B > c_F)^{0.50} \wedge (c_J > c_D)^{0.47}$ | Inhibitory |
| | $\phi_4 = (c_A < 0.84)^{0.76} \wedge (c_H < 1.09)^{0.50}$ | Inhibitory |
| TELLER | [A, F], [C, F], [E, F], [B, F], [D, F] | Excitation |
| OGEM-tab | $[F]$, $[F, A]$ | Excitation |
| | $[A]$ | Inhibitory |

Table 4: Formulas and their effect as learned by CLNN, TELLER and OGEM-tab on company $F$ of LinkedIn.

lustrative example. We select the experimental result on company $F$ of the LinkedIn dataset to demonstrate the expressivity of CLNN's rules, which are shown in Table 4. Here we specify the model to learn five formulas, four of which are inhibitory, and one exhibits excitation. One inhibitory formula has a weight of 0.05, thus not reported in Table 4. Each formula shows the dominant singleton or paired order predicates. Notably, CLNN learns expressive wCL formulas that describe how the logical composition of paired order predicates and(or) singleton order predicates affect a role change in the company $F$. CLNN's rules are more expressive than TELLER and as expressive as OGEM-tab for describing the occurrence of a causal event within a specific historical window.

## 5 CONCLUSION

In this paper, we proposed a novel neuro-symbolic model, CLNN, to learn interpretable wCL formulas from multivariate event data. Experimental results using synthetic and real-world datasets demonstrate CLNN's expressiveness in recovering ground-truth rules in multivariate temporal point processes. Further, CLNN can be trained using gradient-based methods, which improve the learning speed compared to the SOTA.

## 6 ACKNOWLEDGEMENT

This research is sponsored by the Rensselaer-IBM AI Research Collaboration (http://airc.rpi.edu), part of the IBM AI Horizons Network; the National Science Foundation under Grant CMMI-1936578; and the Defense Advanced Research Projects Agency (DARPA) through Cooperative Agreement D20AC00004 awarded by the U.S. Department of the Interior (DOI), Interior Business Center. The content of the information does not necessarily reflect the position or the policy of the Government, and no official endorsement should be inferred.

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

# A  FORMULATION OF LOGICAL CONSTRAINTS & OBJECTIVE FUNCTION

The optimization problem in (10) is formulated by maximizing the log-likelihood subject to the logical constraints for the $\wedge$ and $\vee$ operators. This section discusses the details of the formulation for the two logical constraints and how to formulate the optimization problem while considering the logical constraints. Without loss of generality, we illustrate the formulation of the constraints for the $\wedge$ operator, and the constraints for $\vee$ operator can be derived from the constraints for the $\wedge$ operator using De Morgan's law.

- **Logical constraints for $\wedge$ operator.**

  Let $x, y \in [0, 1]$ denote the inputs of the $\wedge$ operator, and $f(x, y)$ denote the quantitative satisfaction of $\wedge$. The conventional characteristic of the $\wedge$ operator is illustrated as follows: 1) $f(x, y)$ is *low* when either input is *low*, and 2) $f(x, y)$ is *high* when both inputs are *high*. However, we associate each input with a nonnegative weight, implying the input with a zero weight should not affect the output. In other words, if a *low* input has a zero weight, it should not affect the output of $f(x, y)$. Therefore, we require the $\wedge$ operator to exhibit the following characteristics: 1) $f(x, y)$ is *low* when both inputs are *low*, and 2) $f(x, y)$ is *high* when both inputs are *high*. Here we introduce a user-defined hyperparameter $\alpha \in [\frac{1}{2}, 1]$ to capture *low* vs. *high*: $x \in [0, 1 - \alpha)$ represents *low* and $x \in [\alpha, 1]$ represents *high*. According to the above characteristics, we have (Sen et al., 2022)

$$
\begin{aligned}
f(x, y) \leq 1 - \alpha, \quad \forall x, y \in [0, 1 - \alpha), \\
f(x, y) \geq \alpha, \quad \forall x, y \in [\alpha, 1].
\end{aligned}
\tag{13}
$$

  Here we follow a specific choice of $f$ by using a *triangular norm* ($t$-norm) and define the quantitative satisfaction function of $\wedge$ as (Riegel et al., 2020)

$$
p(\mathcal{C}', \phi_1^{w_1} \wedge \phi_2^{w_2}, t) = f(\beta - \sum_{j=1}^{2} w_j(1 - p(\mathcal{C}', \phi_j, t))),
\tag{14}
$$

$$
\text{subject to} \quad \beta - \sum_{j=1}^{2} w_j(1 - \alpha) \geq \alpha, \beta - \sum_{j=1}^{2} w_j \alpha \leq 1 - \alpha,
\tag{15}
$$

  where $f(z) = \max\{0, \min\{z, 1\}\}$ is introduced to clamp the truth value into the range of $[0, 1]$.

- **Logical constraints for $\vee$ operator.**

  By using De Morgan's law, we could derive the quantitative satisfaction function and the logical constraints for the $\vee$ operator with 2 inputs as follows:

$$
p(\mathcal{C}', \phi_1^{w_1} \vee \phi_2^{w_2}, t) = f(1 - \beta + \sum_{j=1}^{2} w_j(p(\mathcal{C}', \phi_j, t))),
\tag{16}
$$

$$
\text{subject to} \quad 1 - \beta + \sum_{j=1}^{2} w_j \alpha \geq \alpha, 1 - \beta + \sum_{j=1}^{2} w_j(1 - \alpha) \leq 1 - \alpha.
\tag{17}
$$

Here we show the characteristics of the activation functions for the $\wedge$ and $\vee$ operators using Figure 4. Figure 4(a) shows the truth value of the $\wedge$ operator with $\alpha = 0.7$. Figure 4(b) shows the truth value of the $\wedge$ operator with $\alpha = 0.9$. It can be distinctly observed that $f(x, y)$ is close to 0 when both $x$ and $y$ are *low*, and $f(x, y)$ is close to 1 when both $x$ and $y$ are *high*. In addition, the unconstrained region for $\alpha = 0.9$ is larger than the unconstrained region for $\alpha = 0.7$. Figure 4(c) shows the truth value of the $\vee$ operator with $\alpha = 0.7$. It is obvious that $f(x, y)$ is close to 0 when both $x$ and $y$ are *low*, and $f(x, y)$ is close to 1 when both $x$ and $y$ are *high*.

In general, we could extend the quantitative satisfaction for the $\wedge$ and $\vee$ operators in (14) - (17) to $k$-ary conjunction and $k$-ary disjunction. The $k$-ary conjunction formulation is expressed as follows.

$$
p(\mathcal{C}', \phi_1^{w_1} \wedge \phi_2^{w_2} \cdots \wedge \phi_k^{w_k}, t) = f(\beta - \sum_{j=1}^{k} w_j(1 - p(\mathcal{C}', \phi_j, t))),
\tag{18}
$$

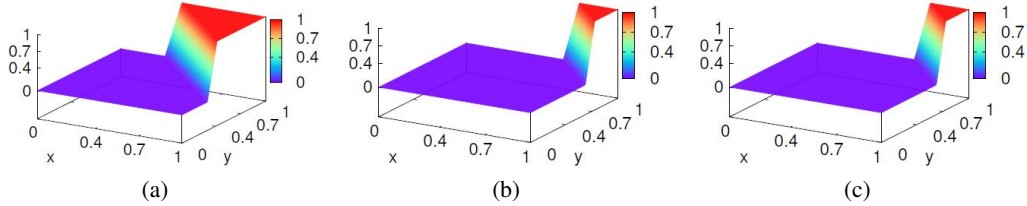

Figure 4: Plot of truth degree for (a) CLNN-$\wedge$ with $\alpha = 0.7$, (b) CLNN-$\wedge$ with $\alpha = 0.9$, (c) CLNN-$\vee$ with $\alpha = 0.7$.

$$\text{subject to} \quad \beta - \sum_{j=1}^{k} w_j(1 - \alpha) \geq \alpha, \beta - \sum_{j=1}^{k} w_j \alpha \leq 1 - \alpha. \tag{19}$$

The $k$-ary disjunction formulation is expressed as follows.

$$p(\mathcal{C}', \phi_1^{w_1} \vee \phi_2^{w_2} \cdots \vee \phi_k^{w_k}, t) \quad = f(1 - \beta + \sum_{j=1}^{k} w_j(p(\mathcal{C}', \phi_j, t))), \tag{20}$$

$$\text{subject to} \quad 1 - \beta + \sum_{j=1}^{k} w_j \alpha \geq \alpha, 1 - \beta + \sum_{j=1}^{k} w_j(1 - \alpha) \leq 1 - \alpha. \tag{21}$$

With the above constraints, we can formulate the maximum likelihood estimation problem as

$$\min \quad -LL_l \tag{22}$$

$$s.t. \quad \forall \phi \in \Phi, \forall 1 \leq k \leq K_\phi^\wedge, \beta_k - \sum_{i \in I_k} w_{i,k}(1 - \alpha) \geq \alpha, \beta_k - \sum_{i \in I_k} w_{i,k} \alpha \leq 1 - \alpha, \tag{23}$$

$$\forall \phi \in \Phi, \forall 1 \leq k' \leq K_\phi^\vee, 1 - \beta_{k'} + \sum_{i \in I_{k'}} w_{i,k'} \alpha \geq \alpha, 1 - \beta_{k'} + \sum_{i \in I_{k'}} w_{i,k'}(1 - \alpha) \leq 1 - \alpha. \tag{24}$$

In this paper, we set $\alpha = 0.5$, thus the constraints in (19) become

$$\sum_{i=1}^{k} w_i \geq 2\beta - 1,$$

$$\sum_{i=1}^{k} w_i \leq 2\beta - 1, \tag{25}$$

$$2\beta - 1 \geq 0,$$
$$w_i \geq 0.$$

Reformulating the above constraints, we have

$$\sum_{i=1}^{k} w_i = 2\beta - 1, \tag{26}$$

$$\beta \geq 0.5,$$
$$w_i \geq 0. \tag{27}$$

The above constraints hold for each conjunction operator in $\phi$. Therefore, we can incorporate the constraints in (26) into the objective function, which becomes

$$\min \quad -LL_l + \sum_{k=1}^{K_\phi^\wedge} (\sum_{i \in I_k} w_{i,k} - 2\beta_k + 1)^2, \tag{28}$$

$$\text{subject to} \quad w_{i,k} \geq 0, \beta_k \geq 0.5, \forall i \in I_k, \forall 1 \leq k \leq K_\phi^\wedge, \forall \phi \in \Phi. \tag{29}$$

Similarly, we propose a set of logical constraints for the $\vee$ operator as (21). If we set $\alpha = 0.5$, the constraints in (21) become

$$\begin{aligned}
\sum_{i=1}^{k} w_i &\geq 2\beta - 1, \\
\sum_{i=1}^{k} w_i &\leq 2\beta - 1, \\
2\beta - 1 &\geq 0, \\
w_i &\geq 0.
\end{aligned} \tag{30}$$

Reformulating the above constraints, we have

$$\sum_{i_1}^{k} w_i = 2\beta - 1, \tag{31}$$

$$\begin{aligned}
\beta &\geq 0.5. \\
w_i &\geq 0.
\end{aligned} \tag{32}$$

The above constraints hold for each disjunction operator in $\phi$. Therefore, we can incorporate the constraints in (31) into the objective function. The maximum likelihood estimation problem then becomes

$$\min \quad -LL_l + \sum_{k=1}^{K_\phi^\wedge} (\sum_{i \in I_k} w_{i,k} - 2\beta_k + 1)^2 + \sum_{k'=1}^{K_\phi^\vee} (\sum_{i \in I_{k'}} w_{i,k'} - 2\beta_{k'} + 1)^2, \tag{33}$$

$$\begin{aligned}
\text{subject to} \quad & w_{i,k} \geq 0, \beta_k \geq 0.5, \forall i \in I_k, \forall 1 \leq k \leq K_\phi^\wedge, \forall \phi \in \Phi, \\
& w_{i,k'} \geq 0, \beta_{k'} \geq 0.5, \forall i \in I_{k'}, \forall 1 \leq k' \leq K_\phi^\vee, \forall \phi \in \Phi.
\end{aligned}$$

## B  PROOF OF THEOREM 8

The activation function designed for the $\wedge$ operator satisfies the properties of nonimpact for zero weights, impact ordering, and monotonicity. Without loss of generality, we present the proof for the $\wedge$ operator connecting two clauses, which can be generalized to the $\wedge$ operator connecting $k$-ary clauses.

**Proof 1** *Here we present the proof for the activation function for the $\wedge$ operator satisfying each property mentioned above.*

- ***Nonimpact for zero weights.***

  *This means if $w_j = 0, j = 1, 2$, then $p(\mathcal{C}', \phi_j, t)$ should have no impact on $p(\mathcal{C}', \phi_1^{w_1} \wedge \phi_2^{w_2}, t)$. Without loss of generality, we suppose $w_1 = 0$, thus we have*

  $$\begin{aligned}
  p(\mathcal{C}', \phi_1^{w_1} \wedge \phi_2^{w_2}, t) &= f(\beta - 0 \cdot (1 - p(\mathcal{C}', \phi_1, t)) - w_2 \cdot (1 - p(\mathcal{C}', \phi_2, t))), \\
  &= f(\beta - w_2 \cdot (1 - p(\mathcal{C}', \phi_2, t))),
  \end{aligned} \tag{34}$$

  *meaning $p(\mathcal{C}', \phi_1, t)$ has no impact on $p(\mathcal{C}', \phi_1^{w_1} \wedge \phi_2^{w_2}, t)$.*

- ***Impact Ordering***

  *This means the truth degree of subformula with higher weights has a greater impact on $p(\mathcal{C}', \phi_1^{w_1} \wedge \phi_2^{w_2}, t)$. Mathematically, we need to prove that if $p(\mathcal{C}', \phi_1, t) = p(\mathcal{C}', \phi_2, t)$ and $w_1 \geq w_2$, then*

  $$\frac{\partial p(\mathcal{C}', \phi_1^{w_1} \wedge \phi_2^{w_2}, t)}{\partial p(\mathcal{C}', \phi_1, t)} \geq \frac{\partial p(\mathcal{C}', \phi_1^{w_1} \wedge \phi_2^{w_2}, t)}{\partial p(\mathcal{C}', \phi_2, t)}. \tag{35}$$

As $f(x) = \max\{0, \min\{x, 1\}\}$, *we have*

$$\frac{df}{dx} = \begin{cases} 0, & \text{if } x < 0, \\ 1, & \text{if } 0 < x < 1, \\ 0, & \text{if } x > 1. \end{cases} \quad (36)$$

*If* $\beta - \sum_{j=1}^{2} w_j(1 - p(\mathcal{C}', \phi_j, t)) < 0$ *or* $\beta - \sum_{j=1}^{2} w_j(1 - p(\mathcal{C}', \phi_j, k)) > 1$*, then we have*

$$\frac{\partial p(\mathcal{C}', \phi_1^{w_1} \wedge \phi_2^{w_2}, t)}{\partial p(\mathcal{C}', \phi_1, t)} = \frac{\partial p(\mathcal{C}', \phi_1^{w_1} \wedge \phi_2^{w_2}, t)}{\partial p(\mathcal{C}', \phi_2, t)} = 0. \quad (37)$$

*Also, if* $0 < \beta - \sum_{j=1}^{2} w_j(1 - p(\mathcal{C}', \phi_j, t)) < 1$*, then we have*

$$\frac{\partial(\beta - \sum_{j=1}^{2} w_j(1 - p(\mathcal{C}', \phi_j, t)))}{\partial p(\mathcal{C}', \phi_1, t)} = w_1(\beta - \sum_{j=1}^{2} w_j(1 - p(\mathcal{C}', \phi_j, t))), \quad (38)$$

*and*

$$\frac{\partial(\beta - \sum_{j=1}^{2} w_j(1 - p(\mathcal{C}', \phi_j, t)))}{\partial p(\mathcal{C}', \phi_2, t)} = w_2(\beta - \sum_{j=1}^{2} w_j(1 - p(\mathcal{C}', \phi_j, t))). \quad (39)$$

*As* $w_1 \geq w_2$*, the following holds:*

$$\frac{\partial p(\mathcal{C}', \phi_1^{w_1} \wedge \phi_2^{w_2}, t)}{\partial p(\mathcal{C}', \phi_1, t)} \geq \frac{\partial p(\mathcal{C}', \phi_1^{w_1} \wedge \phi_2^{w_2}, t)}{\partial p(\mathcal{C}', \phi_2, t)}, \quad (40)$$

*which proves the impact ordering property holds.*

- *Monotonicity.*

  *This means* $p(\mathcal{C}', \phi_1^{w_1} \wedge \phi_2^{w_2}, t)$ *increases monotonically over* $p(\mathcal{C}', \phi_j, t)$*, i.e.*

$$f(\beta - \sum_{j=1}^{2} w_j(1 - p(\mathcal{C}', \phi_j, t))) \leq f(\beta - \sum_{j=1}^{2} w_j(1 - p(\mathcal{C}', \phi_j, t) - d)) \text{ for } d \geq 0. \quad (41)$$

  *First, note that* $\beta - \sum_{j=1}^{2} w_j(1 - p(\mathcal{C}', \phi_j, t))$ *can be rewritten as*

$$\beta - \sum_{j=1}^{2} w_j(1 - p(\mathcal{C}', \phi_j, t)) = \beta - w_1 - w_2 + w_1 p(\mathcal{C}', \phi_1, t) + w_2 p(\mathcal{C}', \phi_2, t). \quad (42)$$

  *This implies* $f(\beta - \sum_{j=1}^{2} w_j(1 - p(\mathcal{C}', \phi_j, t)))$ *is monotonically increasing over* $p(\mathcal{C}', \phi_1, t)$ *and* $p(\mathcal{C}', \phi_2, t)$*. Also, from the proof of impact ordering we know* $f(x) = \max\{0, \min\{x, 1\}\}$ *is monotonically nondecreasing, we can show that*

$$f(\beta - \sum_{j=1}^{2} w_j(1 - p(\mathcal{C}', \phi_j, t))) \leq f(\beta - \sum_{j=1}^{2} w_j(1 - p(\mathcal{C}', \phi_j, t) - d)), d \geq 0. \quad (43)$$

  *Thus the property of monotonicity is satisfied.*

## C  EXPERIMENT RESULTS OF SYNTHETIC DATASETS

**Dataset Generation.** In the experiments on synthetic datasets, we manually generate 3 synthetic datasets considering different settings, where the details and results for the first synthetic dataset is reported in Section 4.2. Each setting considers a different order representation, different number of event labels or different intensity of causal event labels.

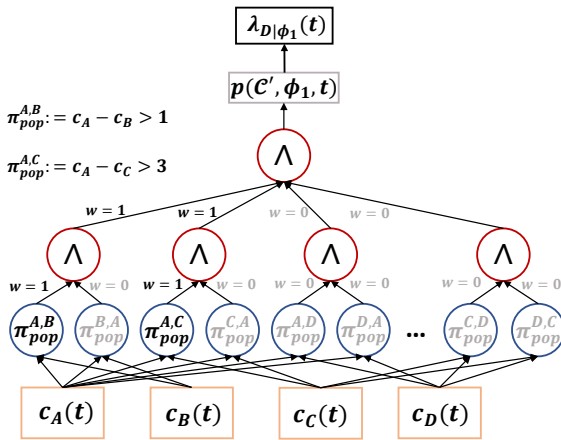

Figure 5: Model structure of $\hat{\phi}_1$ for generating the first synthetic dataset.

## C.1 SYNTHETIC DATASET–1 (SYN-1).

**Generation process.** The first synthetic dataset contains 4 event labels: $A, B, C$, and $D$, where $D$ is the event for prediction, and $A, B, C$ are causal events. The wCL formula used to generate event $D$ in the first synthetic dataset is set as

$$\hat{\phi}_1 = (c_A - c_B > 1)^1 \wedge (c_A - c_C > 3)^1, \tag{44}$$

whose unweighted version reads as "If $A$ happens before $B$ for at least 1 time unit and $A$ happens before $C$ for at least 3 time units, then $D$ will happen."

Here we consider event labels $A, B, C$ as free predicates, whose occurrences are generated by a homogeneous Poisson process. The homogeneous intensity rate for $A, B, C$ are set as $\lambda_A = 0.2$, $\lambda_B = 0.2$, and $\lambda_C = 0.2$. The algorithm used to generate instances of $A, B, C$ is described as Algorithm 1 (Chen, 2016).

---

**Algorithm 1** Simulation of a homogeneous Poisson process with intensity rate $\lambda$.

---

**Input:**
   Intensity rate $\lambda$, simulation horizon $T$
**Output:**
   Occurrence time stamps $\mathcal{T} = \{t_k\}$
1: Initialize $n = 0$, $t_0 = 0$;
2: **while** *True* **do**
3:    Generate $u \sim \text{uniform}(0, 1)$;
4:    Let $w = -ln(u)/\lambda$;
5:    Set $t_{n+1} = t_n + w$;
6:    **if** $t_{n+1} > T$ **then**
7:       **return** $\mathcal{T} = \{t_k\}_{k=1,2,\ldots,n}$;
8:    **else**
9:       Set $n = n + 1$;
10:   **end if**
11: **end while**

---

With the above algorithm, we can generate the occurrences of event labels $A$, $B$, and $C$. Next, we build a CLNN for $\hat{\phi}_1 = (c_A - c_B > 1)^1 \wedge (c_A - c_C > 3)^1$ to calculate the conditional intensity rate $\lambda_{D|\hat{\phi}_1}$, whose model structure is shown in Figure 5. After obtaining $\lambda_{D|\hat{\phi}_1}(t)$, we could use Algorithm 2 (Chen, 2016) to generate the occurrence of $D$.

**Results.** The rules learned by CLNN, TELLER, and OGEM-tab on the first synthetic dataset are presented in Table 5, where the paired order predicate among the two candidates with the highest

---

**Algorithm 2** Simulation of an inhomogeneous Poisson process with intensity rate $\lambda(t)$.

---

**Input:**
   intensity rate $\lambda(t)$, simulation horizon $T$
**Output:**
   Occurrence time stamps $\mathcal{T} = \{t_k\}$
1: Initialize $n = m = 0$, $t_0 = s_0 = 0$, $\bar{\lambda} = \sup_{0 \leq t \leq T}; \lambda(t)$;
2: **while** $s_m < T$ **do**
3:    Generate a uniform random variable $u \sim$ **uniform(0, 1)**;
4:    Let $w = -\ln u / \bar{\lambda}$;
5:    Set $s_{m+1} = s_m + w$;
6:    Generate $D \sim$ **uniform(0,1)**;
7:    **if** $D \leq \lambda(s_{m+1})\bar{\lambda}$ **then**
8:       $t_{n+1} = s_{m+1}$;
9:       $n = n + 1$;
10:   **end if**
11:   $m = m + 1$;
12:   **if** $t_n \leq T$ **then**
13:      **return** $\{t_k\}_{k=1,2,\dots,n}$
14:   **else**
15:      **return** $\{t_k\}_{k=1,2,\dots,n-1}$
16:   **end if**
17: **end while**

---

| Dataset | Syn-1 |
|---|---|
| $N$ (# events) | $N = 4$, $\mathcal{L} = \{A, B, C, D\}$ |
| Ground truth | $\hat{\phi}_1 = (c_A - c_B > 1)^1 \wedge (c_A - c_C > 3)^1$ |
| CLNN's rule | $(c_A - c_B > 1.21)^{1.52} \wedge (c_A - c_C > 3.00)^{1.41} \wedge (c_A - c_D > 0.82)^{0.33} \wedge (c_B - c_C > 4.33)^0 \wedge (c_B - c_D > 10.69)^0 \wedge (c_D - c_C > -6.57)^{0.16}$ |
| TELLER's rule | $A$ before $D$, $B$ before $D$, $C$ before $D$, $A$ before $D$ and $C$ before $D$ |
| OGEM-tab's rule | *Excitation:* $[B]$, $[C]$, $[C, B]$, $[B, C]$, $[A, C, B]$, $[A, B, C]$ 
 *Inhibitory:* $[A]$, $[B, A]$, $[B, A, C]$, $[C, B, A]$, $[A, B]$, $[A, C]$, $[B, C, A]$, $[C, A, B]$, $[C, A]$ |

Table 5: Comparison of rule discovery for CLNN and TELLER on the Syn-1 dataset.

weight is presented. It can be clearly observed that by truncating the predicates with small weights, we could obtain the formula as

$$\phi_1 = (c_A - c_B > 1.21)^{1.52} \wedge (c_A - c_C > 3.00)^{1.41}, \tag{45}$$

which matches well with the ground-truth rule. However, TELLER cannot capture the paired order representation between $A$ and $B$ or $A$ and $C$. OGEM-tab captures the order representation $[A, B]$ and $[A, C]$ as inhibitory causes, which contradicts the ground-truth rule.

### C.2 SYNTHETIC DATASET-2 (SYN-2).

**Generation Process.** The second synthetic dataset contains 5 event labels: $A, B, C, D$ and $E$, where $E$ is the event for prediction, and $A, B, C, D$ are causal events. The wCL formula used to generate the occurrence of event $E$ in the second synthetic dataset is set as

$$\hat{\phi}_2 = (c_A - c_B > 0.5)^1 \wedge (c_A - c_C > 1.5)^1 \wedge (c_C - c_D > 2)^1, \tag{46}$$

whose unweighted version reads as "If $A$ happens before $B$ for at least 0.5 time units, $A$ happens before $C$ for at least 1.5 time units, and $C$ happens before $D$ for at least 2 time units, then E will happen."

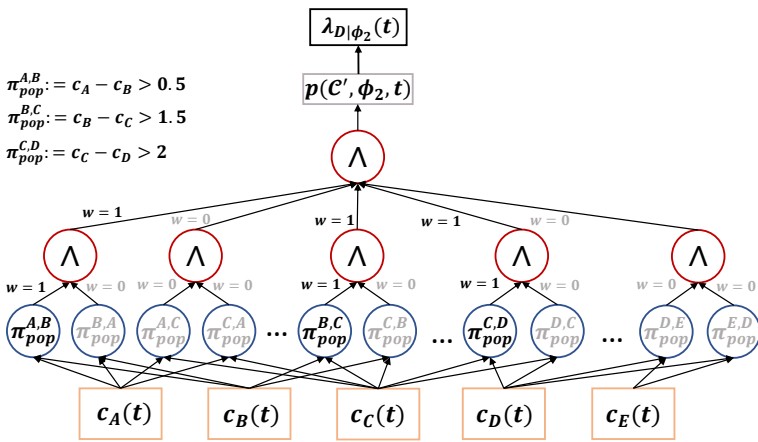

Figure 6: Model structure of $\hat{\phi}_2$ for generating the second synthetic dataset.

The occurrence of events $A, B, C$ and $D$ are generated using Algorithm 1, in which $\lambda_A = \lambda_B = \lambda_C = \lambda_D = 0.2$. After obtaining the occurrence of $A, B, C$ and $D$, we simulate the generation of event label $E$ using Algorithm 2, in which the intensity rate $\lambda_{E|\hat{\phi}_2}(t)$ is computed using the model shown in Figure 6.

**Results.** The rules learned by CLNN, TELLER and OGEM-tab on the second synthetic dataset are presented in Table 6, where the paired order predicate with the highest weight is presented. It can be clearly observed that by truncating the predicates with small weights, CLNN learns a wCL formula as:

$$\phi_2 = (c_A - c_B > 0.77)^{1.27} \wedge (c_A - c_C > 2.09)^{1.15} \wedge (c_C - c_D > 2.60)^{1.06}, \qquad (47)$$

whose order representation match well with the ground-truth rule. Nevertheless, TELLER's rule only capture the ordering between $A$, $B$ and $E$, whereas the ordering between $A$ and $B$ or $B$ and $C$ or $C$ and $D$ are not learned. OGEM-tab's rules can only capture the relation between event label $D$ and event label $E$ can excite the occurrence of event label $E$, whereas not able to capture the dependence of event label $E$'s occurrence on the order relation between $A$ and $B$ or $B$ and $C$ or $C$ and $D$.

| Dataset | Syn-2 |
|---|---|
| $N$ (# events) | $N = 5$, $\mathcal{L} = \{A, B, C, D, E\}$ |
| Ground truth | $\hat{\phi}_2 = (c_A - c_B > 0.5)^1 \wedge (c_B - c_C > 1.5)^1 \wedge (c_C - c_D > 2)^1$ |
| CLNN's rule | $(c_A - c_B > 0.77)^{1.27} \wedge (c_A - c_C > 2.09)^{1.15} \wedge ((c_A - c_D) > -5.00)^{0.25} \wedge ((c_A - c_E) > -2.74)^{0.09} \wedge (c_B - c_C > -9.31)^{0.02} \wedge (c_B - c_D > -8.54)^{0.08} \wedge (c_B - c_E > 2.07)^0 \wedge ((c_C - c_D) > 2.60)^{1.06} \wedge ((c_C - c_E) > -4.27)^{0.03} \wedge ((c_D - c_E) > 1.17)^{0.07}$ |
| TELLER's rule | $A$ before $E$, $B$ before $E$, $A$ and $B$ before $E$, $A$ and $C$ before $E$ |
| OGEM-tab's rule | *Excitation:* $[D]$, $[D, E]$, $[E]$, $[E, D]$ 
 *Inhibitory:* $[D, A]$, $[A]$, $[A, D]$, $[A,D,E]$, $[E, D, A]$, $[D, A, E]$, $[A, E]$, $[E, A]$, $[D, E, A]$, $[A, E, D]$, $[E, A, D]$ |

Table 6: Comparison of rule discovery for CLNN and TELLER on the Syn-2 dataset.

## C.3   SYNTHETIC DATASET 3 (SYN-3).

The third synthetic dataset is generated using a more interesting scheme by combining the generation schemes of the first synthetic dataset and the second synthetic dataset. The third synthetic dataset

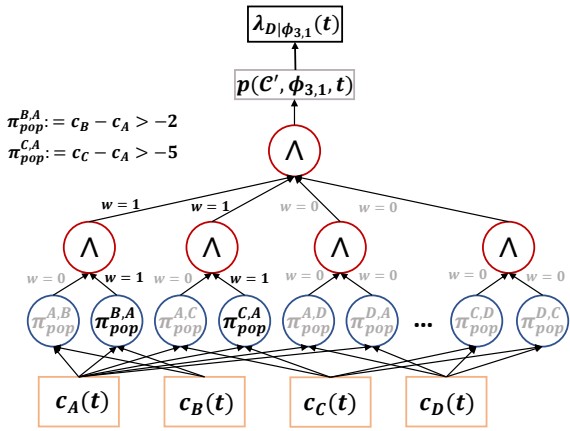

Figure 7: Model structure of $\hat{\phi}_{3,1}$ for generating the occurrence of $D$ in the Syn-3 dataset.

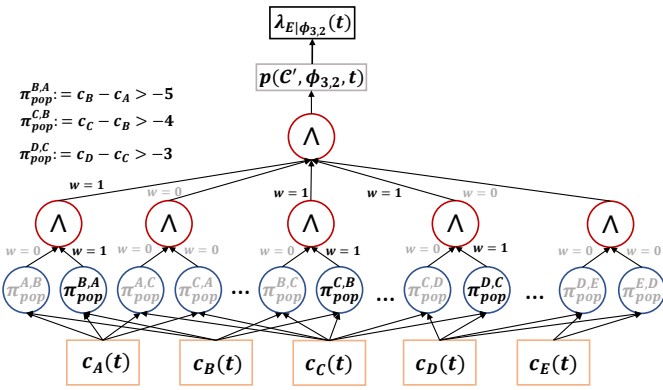

Figure 8: Model structure of $\hat{\phi}_{3,2}$ for generating the occurrence of $E$ in the Syn-3 dataset.

includes five event labels: $A, B, C, D$ and $E$. Here we consider $A, B$, and $C$ as the causal events for the occurrence of $D$, and $A, B, C$, and $D$ as the causal events for the occurrence of $E$. The occurrence of events $A, B, C$ are generated using Algorithm 1, in which $\lambda_A = 0.2$, $\lambda_b = 0.2$, and $\lambda_c = 0.2$. The wCL formula used to generate the occurrence of event $D$ is set as

$$\hat{\phi}_{3,1} = (c_B - c_A > -2)^1 \wedge (c_C - c_A > -5)^1, \tag{48}$$

whose unweighted version reads as "If $A$ happens before $B$ for less than 2 time units, and $A$ happens before $C$ for less than 1 time unit, then D will happen." The generation of $D$'s occurrence follows Algorithm 2, where $\lambda_{D|\hat{\phi}_{3,1}}(t)$ is computed using the model shown in Figure 7. We call the third synthetic dataset at this step as Syn-3.1.

After obtaining the occurrences of events $A, B, C$, and $D$, we could simulate the occurrence of $E$ using the following formula:

$$\hat{\phi}_{3,2} = (c_B - c_A > -5)^1 \wedge (c_C - c_B > -4)^1 \wedge (c_D - c_C > -3)^1. \tag{49}$$

Similarly, the generation of $E$'s occurrence follows Algorithm 2, where the intensity rate $\lambda_{E|\hat{\phi}_{3,2}}(t)$ is computed using the model shown in Figure 8. We call the third synthetic dataset at this step as Syn-3.2.

**Results.**

The rules learned by CLNN, TELLER, and OGEM-tab on the cause of event $D$ in the third synthetic dataset are presented in Table 7, where the paired order predicate with the highest weight among the two candidates is reported. It can be clearly observed that by truncating the predicates with small

weights, CLNN learns a wCL formula as

$$\phi_{3,1} = (c_B - c_A > -1.85)^{1.72} \wedge (c_C - c_A > -3.90)^{1.59}, \tag{50}$$

whose order representation match well with the ground-truth rule. On the other hand, TELLER's rule only reveal the temporal relation between event labels $A$, $B$, $C$ and $D$, but it does not capture the temporal relation between event labels $A$ and $B$ or $A$ and $C$. In addition, we could observe that OGEM-tab does not capture that $C$ is a parent event of $D$.

| Dataset | Syn-3.1 |
|---|---|
| $N$ (# events) | $N = 5, \mathcal{L} = \{A, B, C, D, E\}$ |
| Ground truth | $\hat{\phi}_{3,1} = (c_B - c_A > -2)^1 \wedge (c_C - c_A > -5)^1$ |
| CLNN's rule | $(c_B - c_A > -1.85)^{1.72} \wedge (c_C - c_A > -3.90)^{1.59} \wedge ((c_D - c_A) > -16.25)^{0.33} \wedge ((c_C - c_B) > -3.01)^0 \wedge (c_D - c_B > -7.37)^{0.02} \wedge (c_D - c_C > -7.55)^0$ |
| TELLER's rule | $A$ before $D$, $B$ before $D$, $C$ before $D$ |
| OGEM-tab's rule | *Excitation:* $[A]$, $[A, B, D]$, $[B, D, A]$, $[D, A]$, $[D, A, B]$, $[B, A]$, $[A, D]$, $[D]$, $[B, A, D]$, $[D, B, A]$ 
 *Inhibitory:* $[A, B]$, $[B, D]$, $[B]$, $[A, D, B]$, $[D, B]$ |

Table 7: Comparison of rule discovery of $\phi_{3,1}$ for CLNN and TELLER on the Syn-3.1 dataset.

The rules learned by CLNN, TELLER, and GEM on the cause of event $E$ in the third synthetic dataset are presented in Table 8, in which the discrete wCL formula learned by CLNN is

$$\phi_{3,2} = (c_B - c_A > -3.94)^{1.49} \wedge (c_C - c_B > -3.02)^{2.03} \wedge ((c_D - c_C) > -2.00)^{1.92}. \tag{51}$$

It is obvious that $\phi_{3,2}$ is able to learn the temporal relation between $A$ and $B$, $B$ and $C$, and $C$ and $D$. However, TELLER's rules only reflect the temporal relation between $A, B, C$ and $E$, which cannot give the information about the temporal relation betwee $A$ and $B$, or $B$ and $C$, or $C$ and $D$. OGEM-tab's rule indicate that it considers event labels $A, D, E$ as the parent events of $D$, which does not match with the ground-truth parent set.

| Dataset | Syn-3.2 |
|---|---|
| $N$ (# events) | $N = 5, \mathcal{L} = \{A, B, C, D, E\}$ |
| Ground truth | $\hat{\phi}_{3,2} = (c_B - c_A > -5)^1 \wedge (c_C - c_B > -4)^1 \wedge (c_D - c_C > -3)^1$ |
| CLNN's rule | $(c_B - c_A > -3.94)^{1.49} \wedge (c_C - c_A > -9.12)^{0.25} \wedge ((c_D - c_A) > -1.42)^{0.13} \wedge ((c_E - c_A) > -3.88)^{0.15} \wedge (c_C - c_B > -3.02)^{2.03} \wedge (c_D - c_B > -6.27)^{0.02} \wedge (c_E - c_B > -7.30)^{0.04} \wedge ((c_D - c_C) > -2.00)^{1.92} \wedge ((c_E - c_C) > -5.30)^{0.09} \wedge ((c_E - c_D) > -1.57)^{0.01}$ |
| TELLER's rule | $A$ before $E$, $B$ before $E$, $C$ before $E$ |
| OGEM-tab's rule | *Excitation:* $[A, D]$, $[D, A]$, $[D, E]$, $[E]$, $[A, D, E]$, $[D, E, A]$, $[E, A]$, $[A, E]$, $[E, A, D]$, $[A, E, D]$, $[D, A, E]$, $[E, D, A]$ 
 *Inhibitory:* $[A]$, $[D]$, $[E, D]$ |

Table 8: Comparison of rule discovery of $\phi_{3,2}$ for CLNN and TELLER on the Syn-3.2 dataset.

## C.4 Quantitative Comparison of CLNN's Rules with Ground Truth

To quantitatively evaluate the difference between the ground-truth rules and the rules learned by CLNN, we adopt the Jaccard similarity score to assess the learned formulas against the ground truth. Let $\mathcal{G}$ denote the set of paired ordering representations from the ground-truth rule, and $\mathcal{C}$ denote the set of paired ordering representations from the learned rules, the Jaccard similarity score is calculated as $J = \frac{|\mathcal{C} \cap \mathcal{G}|}{|\mathcal{C} \cup \mathcal{G}|}$. For TELLER and OGEM-tab, the ordering representations are extracted

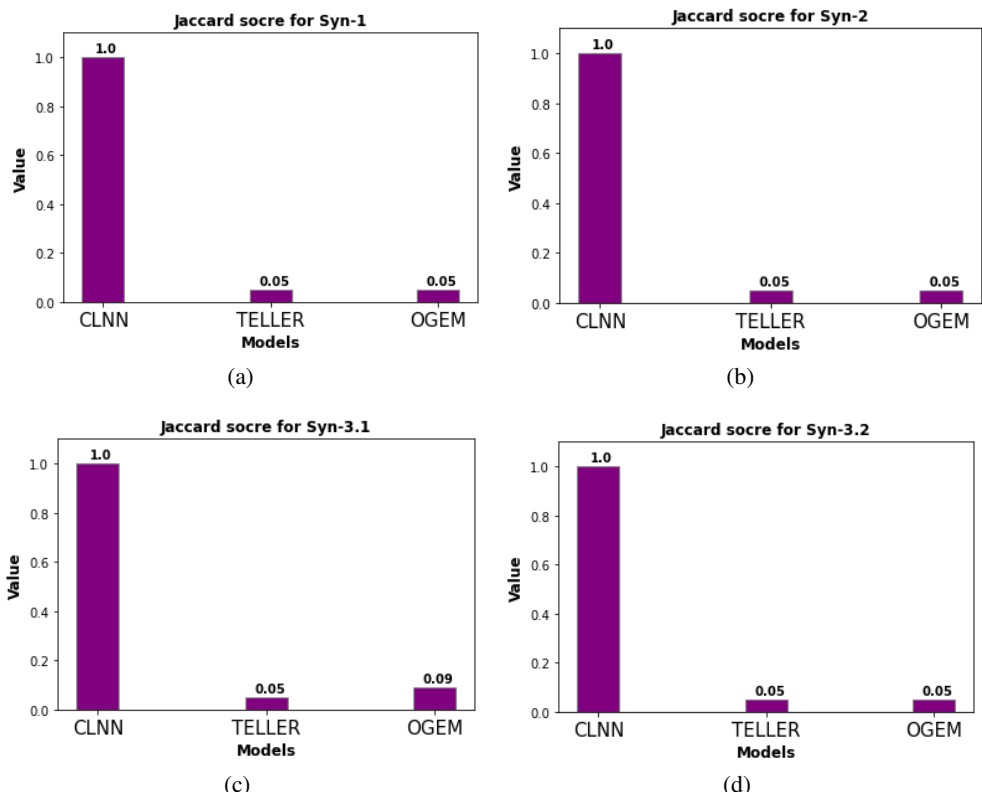

Figure 9: Comparison of ground-truth rules with CLNN's rules in terms of Jaccard similarity score for a) Syn-1, b) Syn-2, c) Syn-3.1, d) Syn-3.2.

from the excitation rules. The comparison of Jaccard similarity score for the synthetic datasets is shown in Figure 9, where the Jaccard similarity score of 0 is manually set to the minimum threshold 0.05 for clarity purposes. It is clearly observed that the Jaccard similarity scores for CLNN is higher than the ones for TELLER or OGEM, implying the rules discovered by CLNN are more consistent with the ground truth.

## C.5 STABILITY ANALYSIS OF CLNN'S RULES WITH RESPECT TO INITIALIZATION

To further validate the model's stability in learning wCL rules, different parameter initialization methods are carried out, including:

1. *rand* – parameter initialization as random numbers from a uniform distribution on the interval $[0, 1)$;

2. *randn* – random numbers from a normal distribution with mean 0 and variance 1;

3. *ones* – constant values of 1;

4. *xavier* – random numbers from a uniform distribution on the interval $[-1/\sqrt{n}, 1/\sqrt{n}]$, where $n$ is the dimension of the parameter.

The rules learned by CLNN for the above parameter initializations are summarized in Table 9. By inspecting the rules for different initialization methods, it is clear that CLNN can still recover the correct paired order representations even if initializing the learning process from a different position. In the meantime, the logic formulas learned by CLNN are stable as the variance of learned parameters is relatively small.

| Dataset | Initialization | Rules |
|---------|---------------|-------|
| Syn - 1 | Ground truth | $\hat{\phi} = (c_A - c_B > 1)^1 \wedge (c_A - c_C > 3)^1$ |
| | $rand$ | $\phi = (c_A - c_B > 1.21)^{1.52} \wedge (c_A - c_C > 3.00)^{1.41}$ |
| | $randn$ | $\phi = (c_A - c_B > 1.21)^{1.58} \wedge (c_A - c_C > 3.32)^{1.56}$ |
| | $ones$ | $\phi = (c_A - c_B > 1.17)^{1.59} \wedge (c_A - c_C > 3.14)^{1.32}$ |
| | $xavier$ | $\phi = (c_A - c_B > 1.12)^{1.45} \wedge (c_A - c_C > 3.20)^{1.33}$ |
| Syn - 2 | Ground truth | $\hat{\phi} = (c_A - c_B > 0.5)^1 \wedge (c_A - c_C > 1.5)^1 \wedge (c_C - c_D > 2)^1$ |
| | $rand$ | $\phi = (c_A - c_B > 0.77)^{1.27} \wedge (c_A - c_C > 2.09)^{1.15} \wedge ((c_C - c_D) > 2.60)^{1.06}$ |
| | $randn$ | $\phi = (c_A - c_B > 0.80)^{1.97} \wedge (c_A - c_C > 1.92)^{1.62} \wedge ((c_C - c_D) > 1.74)^{1.45}$ |
| | $ones$ | $\phi = (c_A - c_B > 1.03)^{1.63} \wedge (c_A - c_C > 1.92)^{1.50} \wedge ((c_C - c_D) > 2.03)^{1.44}$ |
| | $xavier$ | $\phi = (c_A - c_B > 0.97)^{1.92} \wedge (c_A - c_C > 2.07)^{1.63} \wedge ((c_C - c_D) > 1.97)^{1.62}$ |
| Syn - 3.1 | Ground truth | $\hat{\phi} = (c_B - c_A > -2)^1 \wedge (c_C - c_A > -5)^1$ |
| | $rand$ | $\phi = (c_B - c_A > -1.85)^{1.72} \wedge (c_C - c_A > -3.90)^{1.59}$ |
| | $randn$ | $\phi = (c_B - c_A > -1.98)^{1.51} \wedge (c_C - c_A > -3.89)^{1.68}$ |
| | $ones$ | $\phi_{3,1} = (c_B - c_A > -1.94)^{1.84} \wedge (c_C - c_A > -3.68)^{2.33}$ |
| | $xavier$ | $\phi_{3,1} = (c_B - c_A > -1.89)^{1.54} \wedge (c_C - c_A > -3.92)^{1.62}$ |
| Syn - 3.2 | Ground truth | $\hat{\phi} = (c_B - c_A > -5)^1 \wedge (c_C - c_B > -4)^1 \wedge (c_D - c_C > -3)^1$ |
| | $rand$ | $\phi = (c_B - c_A > -3.94)^{1.49} \wedge (c_C - c_B > -3.02)^{2.03} \wedge ((c_D - c_C) > -2.00)^{1.92}$ |
| | $randn$ | $\phi = (c_B - c_A > -3.79)^{1.71} \wedge (c_C - c_B > -3.04)^{1.89} \wedge ((c_D - c_C) > -1.68)^{1.65}$ |
| | $ones$ | $\phi = (c_B - c_A > -3.53)^{1.66} \wedge (c_C - c_B > -3.09)^{1.88} \wedge ((c_D - c_C) > -1.25)^{1.81}$ |
| | $xavier$ | $\phi = (c_B - c_A > -3.71)^{1.53} \wedge (c_C - c_B > -3.09)^{2.04} \wedge ((c_D - c_C) > -1.86)^{1.73}$ |

Table 9: Comparison of rules learned by CLNN for different parameter initialization methods.

### C.6 Analysis of Logical Constraints on the LL

In this part, we investigate the effect of the interpretability using an experiment of the impact of logical constraints on the model's performance. The log-ikelihood on the synthetic datasets for CLNN with and without logical constraints is summarized in Table 10. Table 10 demonstrates that the log-likelihood for CLNN with logical constraints is higher than the log-likelihood for CLNN without constraints, implying that interpretability (logical constraints) is helpful to improve the performance.

| Dataset | CLNN with constraints | CLNN without constraints |
|---------|----------------------|-------------------------|
| Syn - 1 | -7821 | -8716 |
| Syn - 2 | -6075 | -6942 |
| Syn - 3.1 | -10898 | -11583 |
| Syn - 3.2 | -10919 | -11230 |

Table 10: Comparison of LL for CLNN with and without logical constraints.

## D Experiment Results of Real-world Datasets

### D.1 LinkedIn Dataset

The LinkedIn dataset is a collection of job hopping records between 82 IT companies of 3,000 LinkedIn users. Each event stream represents a user's check-in time stamps for different companies or role changes within the same company. Here we select 1000 users' event streams to compose the dataset by filtering out the event streams with uncommon companies, resulting in 10 event labels: $\mathcal{L} = \{A, B, C, D, E, F, G, H, I, J\}$. Here we set the number of formulas as 5, i.e., $\Phi = \{\phi_1, \phi_2, \phi_3, \phi_4, \phi_5\}$, each of which embodies a model structure shown in Figure 2(a) and CLNN aims to learn the parameters for each formula. The weight parameters in the paired order cell or the singleton order cell are initialized as random variables following a Gaussian distribution, and the bias terms of conjunction or disjunction operators are initialized as 1. The architecture weights are initialized as random variables following a Gaussian distribution, and the formula impact weights and bias are initialized as Gaussian random variables. The detailed log-likelihood for each event label is summarized in Table 11.

| Event Label | Log-likelihood |
|:-:|:-:|
| A | -180.59 |
| B | -177.80 |
| C | -89.49 |
| D | -140.31 |
| E | -132.83 |
| F | -76.63 |
| G | -106.23 |
| H | -103.33 |
| I | -95.51 |
| J | -125.45 |

Table 11: Log likelihood for each event label in the LinkedIn dataset.

## D.2 MIMIC II DATASET

MIMIC II dataset is obtained from the intensive care unit research database that consists of 25,328 intensity care unit stays. The records include laboratory data, therapeutic intervention profiles such as nursing progress notes, discharge summaries and others. Here we restrict the event types to the diagnosis of patients and filter out the shorter event sequences with few visits, ending up with 650 patients and 15 event labels: $\mathcal{L} = \{1, 2, 8, 9, 11, 12, 14, 20, 21, 22, 23, 26, 27, 42, 47\}$. Similar to the setting for the LinkedIn dataset, where the initialization of parameters follow the same setting as the LinkedIn dataset. The detailed log-likelihood for each event label is presented in Table 12.

| Event Label | Log-likelihood |
|:-:|:-:|
| 1 | -72.14 |
| 2 | -62.33 |
| 8 | -5.98 |
| 9 | -51.34 |
| 11 | -43.64 |
| 12 | -25.81 |
| 14 | -69.73 |
| 20 | -5.96 |
| 21 | -6.08 |
| 22 | -10.47 |
| 23 | -10.64 |
| 26 | -27.08 |
| 27 | -27.42 |
| 42 | -5.95 |
| 47 | -10.54 |

Table 12: Log likelihood for each event label in the MIMIC II dataset.

## D.3 STACK OVERFLOW DATASET

Stack Overflow is a question-and-answer website spanning a wide range of domains. A badge rewarding scheme is exploited to encourage users to participate in the questioning and answering activities. The badge system of Stack Overflow comprises 81 types of non-topical badges, including the badges that can be awarded only once and the badges that can be awarded several times. The dataset in (Du et al., 2016) was obtained by first filtering out the badges that can be awarded only once, then restricting to the users who have acquired at least 40 badges from 2012-01-01 to 2014-01-01, from which the badges have been awarded more than 100 times are selected as the determinate dataset. Our dataset was acquired by retaining the event streams with one or more of the 20 types of specified badges and then randomly sampling 1000 users to obtain 1000 event streams. The detailed log-likelihood for each event label in the Stack Overflow dataset is summarized in Table 13.

| Event Label | Log-likelihood |
|:-----------:|:--------------:|
| 1 | -3791 |
| 2 | -1451 |
| 3 | -538 |
| 4 | -17656 |
| 5 | -3574 |
| 6 | -3559 |
| 7 | -1381 |
| 8 | -1330 |
| 9 | -10961 |
| 10 | -1105 |
| 11 | -189 |
| 12 | -2012 |
| 13 | -673 |
| 14 | -1340 |
| 15 | -406 |
| 16 | -117 |
| 17 | -186 |
| 18 | -330 |
| 19 | -282 |
| 20 | -100 |

Table 13: Log likelihood for each event label in the Stack Overflow dataset.

| Dataset | CLNN with SOP | CLNN without SOP |
|:-------:|:-------------:|:----------------:|
| LinkedIn | -1228 | -1344 |
| MIMIC II | -436 | -480 |
| Stack Overflow | -50981 | -51195 |

Table 14: Comparison of log-likelihood for CLNN with and without SOP on the real-world datasets.

### D.4 ANALYSIS OF EXPRESSIVENESS ON MODEL'S PERFORMANCE

In this part, we conduct an experiment by training the CLNN without the singleton order cell (SOC) on real-world datasets to show the effectiveness of the singleton order predicates. The comparison of log-likelihood for CLNN with SOC and CLNN without SOC is summarized in Table 14. As evidenced by Table 14, the log-likelihood of CLNN with SOP is higher than the log-likelihood of CLNN without SOP, meaning enriching the expressiveness of wCL formulas can better explain the generative mechanism of events.

