# OpenReview forum: "Weighted Clock Logic Point Process"
_ICLR.cc/2023/Conference — ICLR 2023 poster_

### Official Review · Reviewer_ndLd · 2022-10-28

**Confidence:** 3
**Clarity, Quality, Novelty And Reproducibility:** The paper has good clarity, quality, …
**Correctness:** 4
**Technical Novelty And Significance:** 3
**Empirical Novelty And Significance:** 3
**Recommendation:** 8

**Details Of Ethics Concerns:**

None.

**Strength And Weaknesses:**

Strengths:

- The paper is presented very thoroughly, and the theoretical properties of the model are well established.
- Developing neuro-symbolic approaches is an important problem, both from the point of view of model interpretability, and modeling complex dependencies; the paper offers an elegant solution by formulating a differentiable model which has a logical interpretation.
- The experimentation is somewhat condensed but convincing on both synthetic and real problems.

Weaknesses:

- As above, the experimentation could be expanded.  Mostly, the interpretation is done by simply listing the formulae extracted by the network (Tables 1 and 4) and inviting comparison by inspection.  It would be beneficial to develop a metric to assess the formulae found against the ground truth for various methods.
- The network training incorporates constraints on architecture and parameters (via projected gradient) to ensure interpretability.  It would be interesting to see the log-likelihood performance of similarly sized NN models without these constraints, to gauge if the interpretability is helping or hindering performance.

**Summary Of The Paper:**

The paper introduces a framework of 'clock logic neural networks', which use a special neural network architecture to train temporal process models that can then be interpreted in terms of weighted clock logic formulae.  The paper provides a detailed overview of weighted clock logic, and proves that the network architecture enforces the desired properties necessary for this interpretation.  Training can be achieved directly through projected gradient descent.  The experimentation on synthetic and real datasets shows that the CLNN learns more informative rules (incorporating interval duration information) than alternative SOTA approaches, and achieves competitive performance in terms of the out-of-sample log-likelihood.

**Summary Of The Review:**

An interesting neuro-symbolic approach for training NN models with interpretability in terms of temporal logic.  The paper will be of interest to those working in logic, deep-learning and model interpretability.

---

> ### Author Response · Authors · 2022-11-11
> **Response to Reviewer ndLd**
>
> Thank you for taking the time to review our manuscript and for giving valuable comments. We are grateful that you appreciated the novelty, clarity, and presentation of our work. We have revised the manuscript according to your constructive suggestions. In the following response, we will provide detailed explanations to address each of your concerns.
>
> > __Re. Metric for assessing the formulae found against the ground truth.__
>
> We thank the reviewer for this suggestion. Here we adopt the Jaccard similarity score to assess the learned formulas against the ground truth. Let $\mathcal{G}$ denote the set of paired ordering representations from the ground-truth rule, and $\mathcal{C}$ denote the set of paired ordering representations from the learned rule, the Jaccard similarity score is calculated as $J = \frac{|\mathcal{C}\cap \mathcal{G}|}{|\mathcal{C}\cup \mathcal{G}|}$. The comparison of the Jaccard similarity score for the synthetic datasets is shown in Appendix C.4 of the revised manuscript. It is clearly observed that the Jaccard similarity scores for CLNN are higher than the ones for TELLER or OGEM on all the synthetic datasets, implying the rules discovered by CLNN are more consistent with the ground truth.
>
> > __Re. Log-likelihood for CLNN without constraints.__
>
> We thank the reviewer for this comment. In Appendix C.6 of the revised manuscript, we supplement an experiment to assess the impact of logical constraints on the model's performance. The log-likelihood on the synthetic datasets for CLNN with and without logical constraints is summarized in the table below. The results in the table demonstrate that the log-likelihood for CLNN with logical constraints is higher than the log-likelihood for CLNN without constraints, implying that interpretability (logical constraints) is helpful to improve performance. The table below is also supplemented as Appendix C.6 of the revised manuscript.
>
> Dataset  | CLNN with constraints | CLNN without constraints
> :---:|:---:|:---:
> Syn - 1  | -7821 | -8716
> Syn - 2  | -6075 | -6942
> Syn - 3.1  | -10898 | -11583
> Syn - 3.2 | -10919 | -11230

---

> > ### Comment · Reviewer_ndLd · 2022-12-13
> > **Response**
> >
> > Thanks to the authors for their response.  The incorporation of the comparisons based on the Jaccard similarity and the log-likelihood strengthen the claims of the manuscript.  Particularly, the log-likelihood comparison is valuable for showing that incorporating logical structure improves both interpretability and predictive performance in this context.

---

### Official Review · Reviewer_ryco · 2022-10-29

**Confidence:** 3
**Correctness:** 4
**Technical Novelty And Significance:** 4
**Empirical Novelty And Significance:** 3
**Recommendation:** 8

**Clarity, Quality, Novelty And Reproducibility:**

- The contents of the paper are very technical, but the provided examples are helpful for understanding the high-level intuition behind the different model components.

- To my best knowledge, the proposed approach is novel. References are provided for the building blocks based on earlier works.

- The approach and the experimental setup are described clearly enough to be easily reproduced.

**Strength And Weaknesses:**

Strengths:
- The proposed framework has several clear advantages compared to the existing approaches:
    - The learned logic rules take into account the time between the events (not just their ordering), which can be important in some applications
    - Thanks to using a differentiable relaxation of the discrete search space, the training time for the proposed approach is ~100x faster compared to the existing state-of-the-art method
    - The predictive performance of the proposed model is competitive with other approaches
- The examples introduced throughout the paper are helpful for understanding the contents.
- The experimental setup is described very clearly, which should make the results easy to reproduce.


I haven't found any major weaknesses in the paper. The only part that could be explored in more detail is the stability of the learned logic rules with respect to different random initializations of the network weights and different choices of the  threshold parameter  $\alpha$.

Notes:
- Typo: $c$ should be replaced with $l$ in Equation 9
- Figure 1: Would be helpful to mention that the approach can also work with continuous timestamps, and that discrete timestamps are chosen for illustration purposes.


**Summary Of The Paper:**

This paper proposes an approach for learning interpretable logical rules for event data, where each event is represented by a timestamp and a discrete event type.
The main idea of the proposed approach is to represent the logical rules and their combinations as nodes in a neural network. By optimizing the weights of the neural network with gradient descent, the relevant logical rules are learned from the data. This is a significant improvement compared to earlier works, where the rules are learned using combinatorial optimization tools.

**Summary Of The Review:**

The paper advances the state of the art in the problem of learning interpretable logic rules from multivariate event streams.
The proposed approach offers a clear improvement over existing methods both in terms of expressivity and runtime.

---

> ### Author Response · Authors · 2022-11-11
> **Response to Reviewer ryco**
>
> Thank you for taking the time to review our manuscript and for giving the valuable comments. We are grateful that you appreciated the novelty and advantage of our work. We have revised manuscript and added experiments according to your suggestions. Please kindly find the detailed response to each of your concern as below.
>
> > __Re. The stability of rules with respect to different initialization of the network.__
>
> We thank the reviewer for this comment. To further validate the model's stability in learning wCL rules, four kinds of parameter initialization methods are carried out. Specifically, we name them as 1) _rand_, 2)  _randn_, 3) _ones_, and 4) _xavier_. The details on the parameter initialization and the rules learned by CLNN various initializations are summarized in Appendix C.5 of the revised manuscript. By inspecting the rules for different initialization methods, it is clearly observed that CLNN can still recover the correct paired order representations even if initializing the learning process from a different position. In the meantime, the logic formulas learned by CLNN are stable as the variance of learned parameters is relatively small.
>
> > __Re. Typo: Replacing $c$ with $l$ in Equation 9.__
>
> We thank the reviewer for pointing out this typo. We fixed all the typos and carefully proofread the revised draft.
>
> > __Re. Adding explanation of continuous timestamps to Figure 1.__
>
> We thank the reviewer for this suggestion. In the revised manuscript, we add a remark to the caption in Figure 1 (a) to explain the proposed approach also works for event streams with continuous timestamps.

---

### Official Review · Reviewer_LMp2 · 2022-11-02

**Confidence:** 3
**Correctness:** 3
**Technical Novelty And Significance:** 2
**Empirical Novelty And Significance:** 2
**Recommendation:** 5

**Clarity, Quality, Novelty And Reproducibility:**

- The main challenge with introducing a new formalism to an existing problem is to demonstrate why it is important or useful relative to what exists or could exist.  To the best of my knowledge the approach taken in this paper is sound and well thought through, but it fails in this particular regard.  The first main claim is on expressiveness, which is well supported by the evidence but more context is needed to determine if the gains are meaningful.  For instance, why are performance numbers on the real datasets absent, when these tests were clearly run for the accuracy results?  The second claim is around expressiveness, which as I have noted elsewhere, is barely explored.
- There’s extensive work in various forms of relaxations of propositional formulae and correspondingly learning formulae from a relaxed search space.  The weighted clock logic formalism is new and well-suited to the domain.  In terms of novelty, its relaxation is in line with existing work.
- The writing is locally good.  Individual sentences and paragraphs are clear, and the formalism is well explained.  However, the overall structure could be improved.  For example, the introduction paragraph of Section 2 could provide more context; at the moment one goes through several definitions without a clear direction of where it is headed until the end of the section.  This could be as simple as giving an example of a wCL earlier in the section.
- The mathematical exposition is precise.  On the other hand, to my knowledge, no code has been provided.  Also, optimization hyperparameters and the general experimental set up are mostly absent

**Strength And Weaknesses:**

Strengths:

- Attacks an interesting and useful problem with a well-motivated approach
- The method is performant (in absolute execution time) and has the potential to be a practical tool
- The general approach of relaxation for learning multivariate event streams could generalize to richer representations

Weaknesses

- The experimental results are mixed.  More importantly, it’s not clear when I should expect this method to perform well or poorly.  I would not wish to penalize this paper for not beating SOTA in every situation, but it’s not apparent when and why I should use this approach over any other.  The method performs best on the LinkedIn dataset.  Is this coincidental, or is there some salient property there that makes a difference?
- The primary claimed advantage of CLNNs of SOTA is expressiveness.  While this is to some degree true by construction — CLNNs can represent the relationships between interval lengths and the conditional intensity — there’s not much evidence to demonstrate that it matters in its ability to model a richer class of datasets.

**Summary Of The Paper:**

This paper introduces a method for learning models of multivariate event streams, which are time series of events that may be may occur irregularly and synchronously at points in continuous time.  Specifically, they introduce clock logic neural networks (CLNNs), which model temporal point processes using formulas to represent conditional intensity rates.  Learning in this context involves a relaxation of the satisfiability predicate on formulae and then gradient-based maximum likelihood optimization.

**Summary Of The Review:**

- Well-motivated and interesting problem
- Sound approach but unclear the relevance of the result.  I'm willing to be convinced otherwise
- I’d suggest focusing on the core claims and providing more evidence

---

> ### Author Response · Authors · 2022-11-11
> **Response to Reviewer LMp2**
>
> Thank you for taking the time to review our manuscript and for giving valuable comments. We appreciate the constructive suggestions for our work, which are extremely helpful to improve the quality of this paper. We have revised the manuscript and added experiments according to your comments. Please kindly find our response to each of your concerns below.
>
> > __Re. Model Performance.__
>
> We thank the reviewer for this comment. The benefits of CLNN over the SOTA, TELLER, are mainly two-fold: interpretability and computational cost. On the one hand, CLNN can learn more expressive temporal logic rules than TELLER in that it can not only provide information about the ordering between events but also provide detailed information about the interval length between the ordering events and the occurrence of an event label within a historical window. On the other hand, CLNN is more computationally efficient than TELLER because it learns the temporal logic rules through gradient-based approaches. Therefore, if the user prefers to know how the interval length between ordering events impacts the occurrence of a particular event label and would like to learn the temporal logic rules faster, then CLNN would be an appropriate candidate for such purposes.
>
> The proposed approach first applies masking to the event streams to mask out the duplicated event labels and then retrieves the ordering between event labels. CLNN is more suitable to model event streams such as the LinkedIn dataset where the ordering between events and the interval length between the ordered events have an important impact on the occurrence of a particular event, whereas it might not be suitable to model event streams such as the Stack Overflow dataset where the number of an event label's occurrence has an impact on another event label's occurrence, as the masking function will remove such dependence. If the user has prior knowledge that the number of an event label's occurrences impacts the occurrence of a particular event label, then models such as PGEM are more appropriate for modeling such datasets.
>
> > __Re. The effect of expressiveness in modeling datasets.__
>
> We thank the reviewer for pointing this out. In the revised manuscript, we conduct an experiment by training the CLNN without the singleton order cell (SOC) on the real-world datasets to show the effectiveness of the singleton order predicates. The comparison of log-likelihood for CLNN with SOC and CLNN without SOC is summarized in the table below. As evidenced by the table below, the log-likelihood of CLNN with SOP is higher than the log-likelihood of CLNN without SOP, meaning enriching the expressiveness of wCL formulas can better explain the generative mechanism of events. We include this table as Appendix D.4 of the revised manuscript.
>
>
> | Dataset | CLNN with SOP | CLNN without SOP|
> :---:|:---:|:---:
> |LinkedIn | -1228 | -1344|
> |MIMIC II | -436 | -480|
> |Stack Overflow | -50981 | -51195|
>
> > __Re. Performance numbers on real datasets.__
>
> We thank the reviewer for this comment. We apologize that we did not explain the evaluation on the real-world datasets clearly. As we do not know the ground truth underlying dynamics for real-world datasets, we cannot provide a comparison between the ground-truth rules and the rules learned by CLNN. The methodology of maximizing log-likelihood to model temporal point processes and using log-likelihood as a measure of the model-fit performance is widely accepted (e.g., Bhattacharjya et al., 2020; 2021; Li et al., 2020; 2021 from the manuscript's references). Thus we follow a similar trend to use the log-likelihood to evaluate how well the model fits the event data. In Section 4.3 of the revised manuscript, we report the log-likelihood of CLNN and the SOTA models on the real-world datasets as Table 3.
>
> > __Re. More explanation in Section 2.__
>
> We thank the reviewer for this suggestion. To better explain the terminologies introduced in Section 2, we add a paragraph at the beginning of Section 2.2 to illustrate the overall workflow of the entire framework, accompanied by a visualization of the workflow in Figure 1 (b) of the revised manuscript. In this manner, the terminologies introduced in Section 2 align with the modules in Figure 1 (b) so that the learning process is easier to follow.

---

> > ### Author Response · Authors · 2022-11-11
> > **Response to Reviewer LMp2 (Continued)**
> >
> > > __Re. Providing code and hyperparameters optimization.__
> >
> > We thank the reviewer for this suggestion. We follow the suggestion to supplement a link to the code in Section 4 of the revised manuscript. At the beginning of Section 4 of the revised manuscript, we add a description of the software and hardware utilized for the experiments. In Section 4.2 of the revised manuscript, we add the experimental setup for the experiments on the synthetic datasets, including the pre-specified parameters for generating the synthetic data, the truth value threshold, and the parameter initialization. In Section 4.3 of the revised manuscript, we add the experimental setup for the experiments on the real-world datasets, including the truth value threshold, the number of subformulas, and the parameter initialization. In addition, we performed studies on the hyperparameters setting, specifically investigating the impact of different initialization methods on the rules learned by CLNN. The experimental results for the hyperparameter testing are shown in Appendix C.5 of the revised manuscript.

---

### Official Review · Reviewer_7kNw · 2022-11-03

**Confidence:** 3
**Clarity, Quality, Novelty And Reproducibility:** The writing is clear and the paper sh…
**Correctness:** 3
**Technical Novelty And Significance:** 2
**Empirical Novelty And Significance:** 2
**Recommendation:** 5

**Strength And Weaknesses:**

The approach of using neural networks to model the time interval length is novel. The writing of this paper, especially the technical part, is sound and with minor issues. There are some insights of how to relax logical formulas that people can draw from this paper.

The major concern of this paper is that there are no theoretical insights or motivations for the proposed method. The authors present the architecture of the clock logic neural networks without motivating the reason behind them, nor do they explain how the smooth relaxation or POC/SOC benefits problem-solving, either in the method section or in the experiments section.

In section 2, though the examples are easy to follow, the authors introduce more than enough terminologies which increases the complexity of understanding. For example, it seems only "paired order predicate" is used in the rest of this paper so other terminologies can be simplified.

The author claims to learn "interpretable" rules without detailed explanation/verification, either in the main method section or in the experiments section.

The synthetic dataset is too simple and thus easy to learn. The "effectiveness" of the proposed method needs to be verified on larger datasets or real-world datasets. The empirical results on real-world datasets are not competitive enough compared to existing methods.



**Summary Of The Paper:**

This paper studies event streams and proposes a neural-network-based method that learns weighted clock logic to model temporal point process. Compared to traditional combinatorial optimization approaches, this method is more expressive in terms of representing weighted clock logic formulas as neural network activation function; easier to learn through gradient descent.

**Summary Of The Review:**

This paper proposes a novel approach to learn temporal point processes using neural networks which learn weighted clock logic formulas. Though the presentation is clear, the major concerns of this paper are the lack of motivation and competitive empirical results. I am not an expert on this topic, but I think the authors can improve this paper by providing clear motivation/theoretical insights, or detailed explanation/verification of all the claims (such as interpretability and effectiveness).

---

> ### Author Response · Authors · 2022-11-11
> **Response to Reviewer 7kNw**
>
> Thank you for taking the time to review our manuscript and for giving nice comments and constructive suggestions. All your suggestions will help us to further improve the quality of this paper. We have revised the manuscript and added more explanations according to your comments. In the following response, we will provide detailed explanations to address each of your concerns.
>
>
> > __Re. Motivation.__
>
> We thank the reviewer for this comment. The motivation for the clock logic neural network is mainly two-fold: interpretability and computational efficiency. From the perspective of interpretability, the SOTA work, TELLER, can only learn rules about the ordering between events, whereas the time interval between the ordered events cannot be captured. To offset this limitation, we propose POPs and SOPs with learnable parameters for capturing such missing information. From the perspective of computational efficiency, TELLER learns temporal logic rules by searching for effective rules from the exponentially large collection of all possible rules through a branch-and-price algorithm with a high computational cost. To tackle such limitations, we design smooth activation functions for POP, SOP, and the logical operators so that the parameters of wCL formulas can be learned through maximum likelihood estimation using gradient-based methods. The above design allows the proposed architecture of CLNN to efficiently learn more expressive formulas than TELLER. In Section 1 and 2.3 of the revised manuscript, we add the motivation for the design of POP and SOP. Below equations (4) and (6) of the revised manuscript, we add the benefits for the smooth design of the activation functions for the POP, SOP, and the logical operators so that the motivation is clearer.
>
> > __Re. Simplifying terminology definition in Section 2.__
>
> We thank the reviewer for this comment. We apologize that we did not clearly explain that the POPs and the SOPs are both used in the design of CLNN and the experiment section. For clarity of the usage of the terminologies introduced in Section 2, we add a paragraph at the beginning of Section 2.2 to illustrate the overall workflow of the proposed framework, accompanied by a visualization of the workflow in Figure 1 (b) of the revised manuscript. In this manner, the terminologies introduced in Section 2 align with the modules in Figure 1 (b) so that the learning process is easier to follow.
>
> > __Re. Explanation/verification of "interpretable rules".__
>
> We thank the reviewer for this comment. In the revised manuscript, we add the explanation and verification for the interpretability of the learned rules using Example 1 below Remark 7 and Table 1 in Section 4.2. The wCL formula shown in Example 1, $\phi = (c_A-c_B>1)^1\wedge (c_C< 3)^{0.05}$, is human-readable and interpretable in that the first and second clauses read “$A$ happened before $B$ for at least one day'' and “$C$ happened less than 3 days ago'', respectively. Table 1 in the revised manuscript shows the wCL formula learned by CLNN on the first synthetic dataset, expressed as $\phi = (c_A-c_B>1.21)^{1.52} \wedge (c_A-c_C>3.00)^{1.41}$. Here $\phi$ can be interpreted as “If $A$ happens before $B$ for at least $1.21$ time units and $A$ happens before $C$ for at least $3.00$ time units, then $D$ will happen''. In addition, we add a Jaccard similarity score in Appendix C.4 of the revised manuscript to evaluate the similarity between the ground-truth rules and the rules learned by CLNN. The Jaccard similarity score demonstrates that CLNN's rules are more consistent with the ground-truth rules than the other models.
>
> > __Re. Performance on synthetic and real-world datasets.__
>
> We thank the reviewer for this comment. Learning rules from these synthetic datasets is actually not an easy task, which can be observed from the experimental results of TELLER and OGEM. The rules learned by TELLER and OGEM are different from the ground-truth rules, but the rules learned by CLNN match well with the ground truth, as evidenced by the Jaccard score supplemented in Appendix C.4 of the revised manuscript.
>
> CLNN is more suitable to model event streams  such as the LinkedIn dataset where the ordering between events and the interval length between the ordered events have an impact on the occurrence of a particular event label. However, it might not be suitable to model event streams such as the Stack Overflow dataset, where the number of an event label's occurrence has an impact on the occurrence of a particular event, as the masking function will remove such dependence. CLNN would be a more suitable choice if one prefers to know how the interval length between ordering events impacts another event label's occurrence or would like to learn the temporal logic rules faster.

---

### Author Response · Authors · 2022-11-11
**Response to all reviewers**

The authors would like to thank the reviewers for taking the time to review our manuscript and for providing constructive feedback.

We have uploaded a revised manuscript which has been modified according to the reviewers' comments and suggestions, with the revisions highlighted in blue. We summarize the major revisions below, and more details can be found in the responses to individual reviewers. All the revisions mentioned in this response are referred to as the revisions in the revised manuscript.

* We add more details on the motivation of the proposed model in the Introduction and Methodology sections of the revised manuscript so that the contribution of our paper is clearer.
* We add a paragraph at the beginning of Section 2.2 of the revised manuscript to explain the overall workflow of the proposed model, with Figure 1 (b) of the revised manuscript as an illustration.
* We add more explanation on the rules that can be learned by CLNN to show the interpretability in Section 2.2, 3.2, and 4.2 of the revised manuscript.
* In Appendix D.4 of the revised manuscript, we add an experiment on real-world datasets to show the impact of expressiveness on modeling the datasets.
* In Section 4.3 of the revised manuscript, we add more explanation on using the log-likelihood as a metric to evaluate the performance of the models.
* In Appendix C.5 of the revised manuscript, we add an experiment to explore the stability of learned logic rules with respect to different random initializations of the model.
* In Appendix C.4 of the revised manuscript, we add a quantitative metric to assess the similarity between the formulas learned by various models and the ground-truth rules.
* In Appendix C.6 of the revised manuscript, we add an experiment to assess the effect of interpretability on the model's performance.

Efforts were also made to correct the mistakes and improve the writing of the manuscript.

---

### Decision · Program_Chairs · 2023-01-20

**Decision:**

Accept: poster

**Justification For Why Not Higher Score:**

Empirical results are not strong enough for a higher score.

**Justification For Why Not Lower Score:**

It's a thorough interesting paper, certainly something that can be built on.

**Metareview: Summary, Strengths And Weaknesses:**

This paper proposes a neuro-symbolic model for continuous event streams. Reviewers appreciate the proposed model but had some reservations about the empirical results, which are sufficiently addressed in the author response.

**Note From Pc:**

if the above contains the word "oral" or "spotlight" please see: "oral" presentation means -> notable-top-5% and "spotlight" means -> notable-top-25%. As stated in our emails, we are disassociating presentation type from AC recommendations